# CCR5 deficiency impairs CD4+ T-cell memory responses and antigenic sensitivity through increased ceramide synthesis

Ana Martín-Leal[1,†], Raquel Blanco[1,†], Josefina Casas[2,3], María E Sáez[4], Elena Rodríguez-Bovolenta[5], Itziar de Rojas[6], Carina Drechsler[7,8,9], Luis Miguel Real[10,11], Gemma Fabrias[2,3], Agustín Ruíz[6,12], Mario Castro[13] (ID), Wolfgang WA Schamel[7,8,14] (ID), Balbino Alarcón[5] (ID), Hisse M van Santen[5] (ID) & Santos Mañes[1,*] (ID)

## Abstract

CCR5 is not only a coreceptor for HIV-1 infection in CD4+ T cells, but also contributes to their functional fitness. Here, we show that by limiting transcription of specific ceramide synthases, CCR5 signaling reduces ceramide levels and thereby increases T-cell antigen receptor (TCR) nanoclustering in antigen-experienced mouse and human CD4+ T cells. This activity is CCR5-specific and independent of CCR5 co-stimulatory activity. CCR5-deficient mice showed reduced production of high-affinity class-switched antibodies, but only after antigen rechallenge, which implies an impaired memory CD4+ T-cell response. This study identifies a CCR5 function in the generation of CD4+ T-cell memory responses and establishes an antigen-independent mechanism that regulates TCR nanoclustering by altering specific lipid species.

Keywords  ccr5[delta]32; humoral response; membrane phase; sphingolipid; T-cell receptor

Subject Category  Immunology

The EMBO Journal (2020) 39: e104749

See also: **C Matti & DF Legler** (August 2020)

## Introduction

The C-C motif chemokine receptor 5 (CCR5) is a seven-transmembrane G protein-coupled receptor (GPCR) expressed on the surface of several innate and adaptive immune cell subtypes, including effector and memory CD4+ T lymphocytes (Gonzalez-Martin *et al*, 2012). CCR5 acts also a necessary coreceptor for infection by HIV-1. An HIV-resistant population served to identify a 32-bp deletion within the CCR5 coding region (*ccr5Δ32*), which yields a non-functional receptor (Blanpain *et al*, 2002). Since *ccr5Δ32* homozygous individuals are seemingly healthy, a radical body of thought considers that CCR5 is dispensable for immune cell function.

Experimental and epidemiological evidence nonetheless indicates that CCR5 has an important role in innate and acquired immune responses. CCR5 and its ligands C-C motif ligand 3 (CCL3; also termed macrophage inflammatory protein [MIP]-1α), CCL4 (MIP-1β), CCL5 (regulated upon activation, normal T cell expressed and secreted [RANTES]), and CCL3L1 have been associated with exacerbation of chronic inflammatory and autoimmune diseases. Despite varying information due probably to ethnicity effects (Lee *et al*, 2013; Schauren *et al*, 2013), further complicated in admixed populations (Toson *et al*, 2017), epidemiological studies support the

1  Department of Immunology and Oncology, Centro Nacional de Biotecnología (CNB/CSIC), Madrid, Spain
2  Department of Biological Chemistry, Institute of Advanced Chemistry of Catalonia (IQAC-CSIC), Barcelona, Spain
3  CIBER Liver and Digestive Diseases (CIBER-EDH), Instituto de Salud Carlos III, Madrid, Spain
4  Centro Andaluz de Estudios Bioinformáticos (CAEBi), Seville, Spain
5  Department of Cell Biology and Immunology, Centro de Biología Molecular Severo Ochoa (CBMSO/CSIC), Madrid, Spain
6  Alzheimer Research Center, Memory Clinic of the Fundació ACE, Institut Català de Neurociències Aplicades, Barcelona, Spain
7  Signaling Research Centers BIOSS and CIBSS, University of Freiburg, Freiburg, Germany
8  Department of Immunology, Faculty of Biology, University of Freiburg, Freiburg, Germany
9  Institute for Pharmaceutical Sciences, University of Freiburg, Freiburg, Germany
10  Unit of Infectious Diseases and Microbiology, Hospital Universitario de Valme, Seville, Spain
11  Department of Biochemistry, Molecular Biology and Immunology, School of Medicine, Universidad de Málaga, Málaga, Spain
12  CIBER Enfermedades Neurodegenerativas (CIBERNED), Instituto de Salud Carlos III, Madrid, Spain
13  Interdisciplinary Group of Complex Systems, Escuela Técnica Superior de Ingeniería, Universidad Pontificia Comillas, Madrid, Spain
14  Centre for Chronic Immunodeficiency (CCI), University of Freiburg, Freiburg, Germany
  *Corresponding author. Tel: +34 91 585 4840; Fax: +34 91 372 0493; E-mail: smanes@cnb.csic.es
  †These authors contributed equally to this work.

ccr5Δ32 allele as a marker for good prognosis for these overreactive immune diseases (Vangelista & Vento, 2017). In contrast, ccr5Δ32 homozygotes are prone to fatal infections by several pathogens such as influenza, West Nile, and tick-borne encephalitis viruses (Lim & Murphy, 2011; Falcon et al, 2015; Ellwanger & Chies, 2019). The mechanisms by which the ccr5Δ32 polymorphism affects all these pathologies have usually been linked to the capacity of CCR5 to regulate leukocyte trafficking. For example, CCR5 deficiency reduces recruitment of influenza-specific memory CD8[+] T cells and accelerates macrophage accumulation in lung airways during virus rechallenge (Dawson et al, 2000; Kohlmeier et al, 2008); this could lead to acute severe pneumonitis, a fatal flu complication. CCR5 nonetheless has migration-independent functions that maximize T-cell activation by affecting immunological synapse (IS) formation (Molon et al, 2005; Floto et al, 2006; Franciszkiewicz et al, 2009) as well as T-cell transcription programs associated with cytokine production (Lillard et al, 2001; Camargo et al, 2009). CCR5 and its ligands are also critical for cell-mediated immunity to tumors and pathogens, including HIV-1 (Dolan et al, 2007; Ugurel et al, 2008; González-Martín et al, 2011; Bedognetti et al, 2013).

Whereas the role of CCR5 in T-cell priming is well established, its involvement in memory responses has not been addressed in depth. Only a single report suggested CCR5 involvement in CD4[+] T-cell promotion of memory CD8[+] T-cell generation through a migration-dependent process (Castellino et al, 2006). It remains unknown whether CCR5 endows memory T cells with additional properties. One such property is the elevated sensitivity of effector and memory ("antigen-experienced") CD4[+] and CD8[+] T cells to their cognate antigen compared to naïve cells (Kimachi et al, 1997; Kersh et al, 2003; Huang et al, 2013). This sensitivity gradient (memory ≫ effector > naïve) in CD8[+] T cells is linked to increased valency of preformed T-cell antigen receptor (TCR) oligomers at the cell surface, termed TCR nanoclusters (Kumar et al, 2011). This antigen-independent TCR nanoclustering (Schamel et al, 2005, 2006; Lillemeier et al, 2010; Sherman et al, 2011; Schamel & Alarcon, 2013) enhances antigenic sensitivity by increasing avidity to multimeric peptide-major histocompatibility complexes (Kumar et al, 2011; Molnar et al, 2012) and by allowing cooperativity between TCR molecules (Martínez-Martín et al, 2009; Martín-Blanco et al, 2018). TCRβ subunit interaction with cholesterol (Chol) and the presence of sphingomyelins (SM) are both essential for TCR nanoclustering (Molnar et al, 2012; Beck-Garcia et al, 2015). Replacement of Chol by Chol sulfate impedes TCR nanocluster formation and reduces CD4[+]CD8[+] thymocyte sensitivity to weak antigenic peptides (Wang et al, 2016). Whether antigen-experienced CD4[+] T-cell sensitivity is linked to TCR nanoscopic organization and the homeostatic factors that regulate TCR nanoclustering remains unexplored.

Given its co-stimulatory role in CD4[+] T cells, we speculated that CCR5 signals would affect the antigenic sensitivity of CD4[+] memory T cells. To test this hypothesis, we analyzed the function of in vivo-generated memory CD4[+] T cells in wild-type (WT) and CCR5[−/−] mice, and the effect of CCR5 deficiency on CD4 T-cell help in the T-dependent humoral response. We found that CCR5 is necessary for the establishment of a functional CD4 memory response through a mechanism independent of its co-stimulatory role for the TCR signal. We show that CCR5 deficiency does not affect memory CD4 T-cell generation, but reduces their sensitivity to antigen. Our data demonstrate an unreported CCR5 regulatory role in memory CD4[+]

T-cell function by inhibiting the synthesis of ceramides, which are identified here as negative membrane regulators of TCR nanoscopic organization.

# Results

## CCR5 deficiency impairs the CD4[+] T-cell memory response

To determine the role of CCR5 in CD4[+] memory T-cell generation and/or function, we adoptively transferred congenic CD45.1 mice with lymph node/spleen cell suspensions from OT-II WT or CCR5[−/−] mice (CD45.2) and subsequently infected them with OVA-encoding vaccinia virus; 5 weeks post-immunization, we analyzed spleen CD45.2[+] donor cells from OT-II mice. CCR5 expression on OT-II cells affected neither the total number of memory CD4[+] T cells (Fig 1A and B) nor the percentage of CD4[+] $T_{EM}$ (CD44[hi]; CD62L[−]; Fig 1C) or $T_{CM}$ (CD44[hi]; CD62L[+]; Fig 1D) cells generated. OT-II WT cells nonetheless had stronger responses to antigenic restimulation than OT-II CCR5[−/−] memory T cells, as determined by the percentage of interferon (IFN)γ-producing cells after ex vivo stimulation with OVA_{323–339} (Fig 1E).

We also studied T cell-dependent B-cell responses in WT and CCR5[−/−] mice after immunization with the hapten 4-hydroxy-3-iodo-5-nitrophenylacetyl coupled to ovalbumin (NIP-OVA; Fig 1F). We detected no difference in the percentage or absolute number of T follicular helper ($T_{fh}$) cells (CD4[+], CD44[hi], CXCR5[+], PD1[+]) between WT and CCR5[−/−] mice at 7 days post-immunization (Fig 1G–I). At day 30, half of the mice were boosted with the same NIP-OVA immunogen (OVA/OVA) and the other half received NIP conjugated with another carrier protein (OVA/KLH); levels of NIP-specific high- and low-affinity immunoglobulins (Ig) were analyzed 15 days later. Comparison of the humoral responses between OVA/OVA- and OVA/KLH-immunized mice would assess the effect of memory CD4[+] T cells specific for the first carrier protein on the humoral response to NIP. There were no differences in high/low-affinity NIP-specific IgM production between WT and CCR5[−/−] mice with either immunization strategy (Fig 1J and K). CCR5 deficiency markedly impaired the generation of high-affinity class-switched anti-NIP antibodies specifically in OVA/OVA-immunized mice (Fig 1J and K). Since class switching was similar in WT and CCR5[−/−] OVA/KLH-immunized mice, our results suggest that CCR5 deficiency reduces the generation of high-affinity class-switched immunoglobulins due to deficient memory CD4[+] T-cell function.

## The CCR5 effect on antigen-experienced CD4[+] T cells is cell-autonomous

To test whether the in vivo memory defect associated with CCR5 deficiency was intrinsic to CD4[+] T cells, we activated OT-II WT and CCR5[−/−] spleen T cells with OVA_{323–339} antigen for 3 days; after antigen removal, we cultured cells with IL-2 or IL-15. OT-II cells that differentiated in exogenous IL-2 expressed CCL3, CCL4, CCL5, and a functional CCR5 receptor, as determined by their ability to flux Ca[2+] and migrate after CCL4 stimulation (Appendix Fig S1A–D).

Like CD8[+] T cells (Richer et al, 2015), OT-II cells cultured with IL-15 showed a memory-like phenotype (Fig EV1); they were smaller than IL-2-cultured cells and retained CD62L with reduced

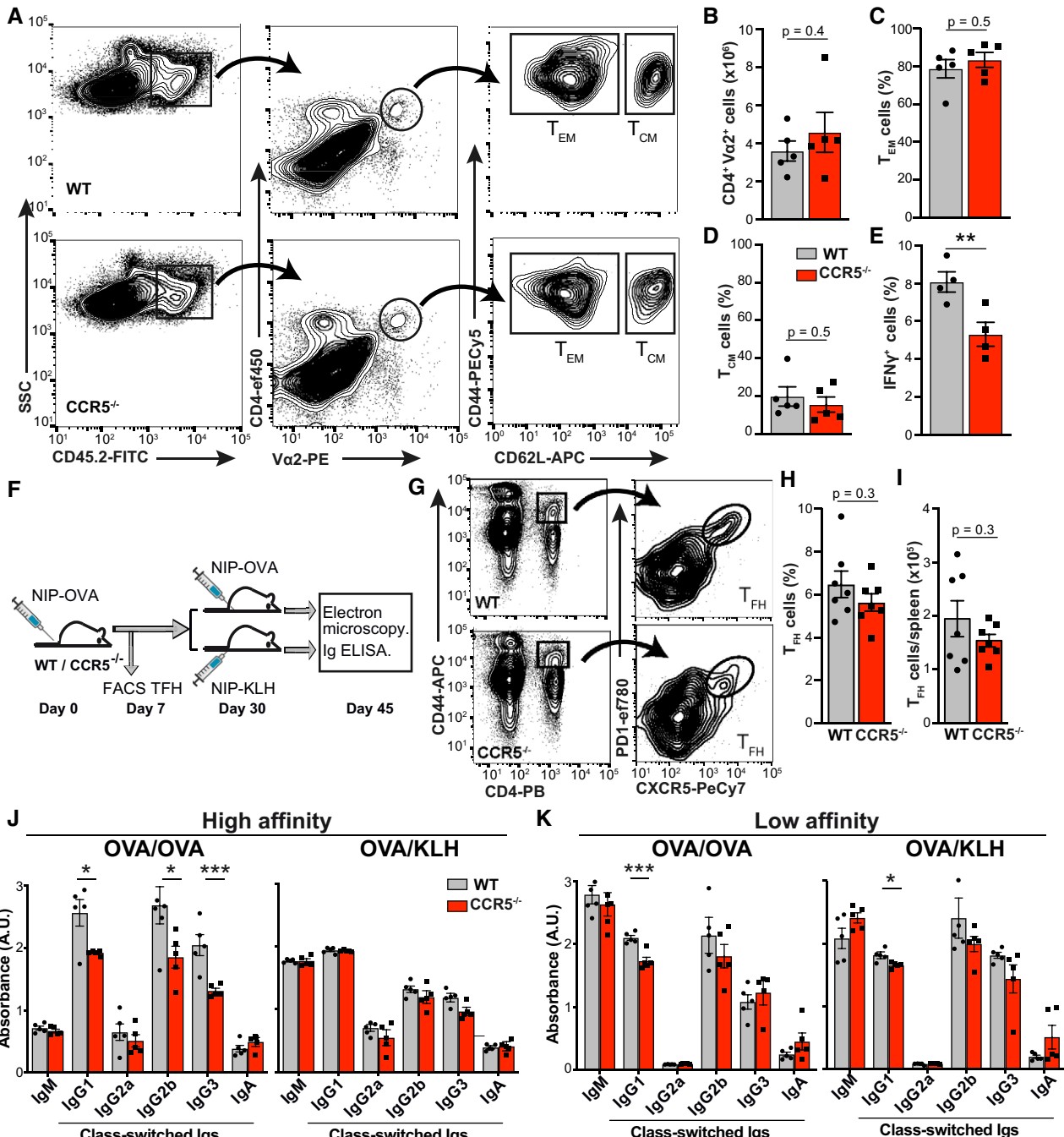

**Figure 1. CCR5 deficiency impairs CD4+ T-cell memory responses.**

A   Representative plots of splenocytes from CD45.1 mice adoptively transferred with CD45.2 OT-II WT or CCR5$^{-/-}$ lymph node cell suspensions, 5 weeks after infection with rVACV-OVA virus. The gating strategy used to identify the memory CD4+ T-cell subtypes is shown ($n = 5$).

B   Absolute number of OT-II cells recovered in spleens of mice as in A ($n = 5$).

C, D   Percentage of CD4+ $T_{EM}$ (C) and $T_{CM}$ (D) in the OT-II WT and CCR5$^{-/-}$ populations ($n = 5$).

E   IFNγ-producing OT-II WT and CCR5$^{-/-}$ memory cells isolated from mice as in (A) and restimulated *ex vivo* with OVA$_{323-339}$ (1 μM) ($n = 4$).

F   Immunization scheme for NIP-OVA and NIP-KLH in WT and CCR5$^{-/-}$ mice.

G–I   Representative plots (G) and quantification of the frequency (H) and absolute number (I) of $T_{fh}$ cells (CD4+CD44+PD-1+CXCR5+) in the spleen after primary immunization (day 7) with NIP-OVA ($n = 7$).

J, K   ELISA analysis of high- (J) and low-affinity (K) isotype-specific anti-NIP antibodies in sera from OVA/OVA- and OVA/KLH-immunized mice (day 15 post-challenge; $n = 5$ mice/group). Data representative of one experiment of two.

Data information: (B–E, H–K), Data are mean ± SEM. *$P < 0.05$, **$P < 0.01$, ***$P < 0.001$, two-tailed unpaired Student's *t*-test.

activation marker expression (CD25, CD69, CD44) compared to IL-2-cultured T cells (Fig 2A). Findings were similar in OT-II WT and CCR5$^{-/-}$ cells (Fig 2B), which reinforced the idea that CCR5 is not involved in CD4$^+$ T memory cell differentiation. Restimulation of IL-2- or IL-15-expanded OT-II lymphoblasts with the OVA$_{323-339}$ peptide nonetheless indicated that CCR5-expressing cells showed strong proliferation and higher IL-2 production at low antigen concentrations than CCR5-deficient cells (Fig 2C–F), indicative of an increased number of cells responding to antigenic stimulation. CCR5 might thus increase the antigenic sensitivity of antigen-experienced CD4$^+$ T cells in a cell-autonomous manner.

### CCR5 modulates TCR nanoclustering in antigen-experienced CD4$^+$ T cells

The high antigenic sensitivity of antigen-experienced CD8$^+$ T cells was partially attributed to increased TCR nanoclustering (Kumar *et al*, 2011). To determine whether CCR5 deficiency influences TCR organization, we used electron microscopy (EM) to analyze surface replicas of OT-II WT and CCR5$^{-/-}$ naïve cells and lymphoblasts after labeling with anti-CD3ε antibody and 10 nm gold-conjugated protein A; a representative image of a IL-15-expanded WT lymphoblast is shown (Fig EV2). We found no differences in TCR

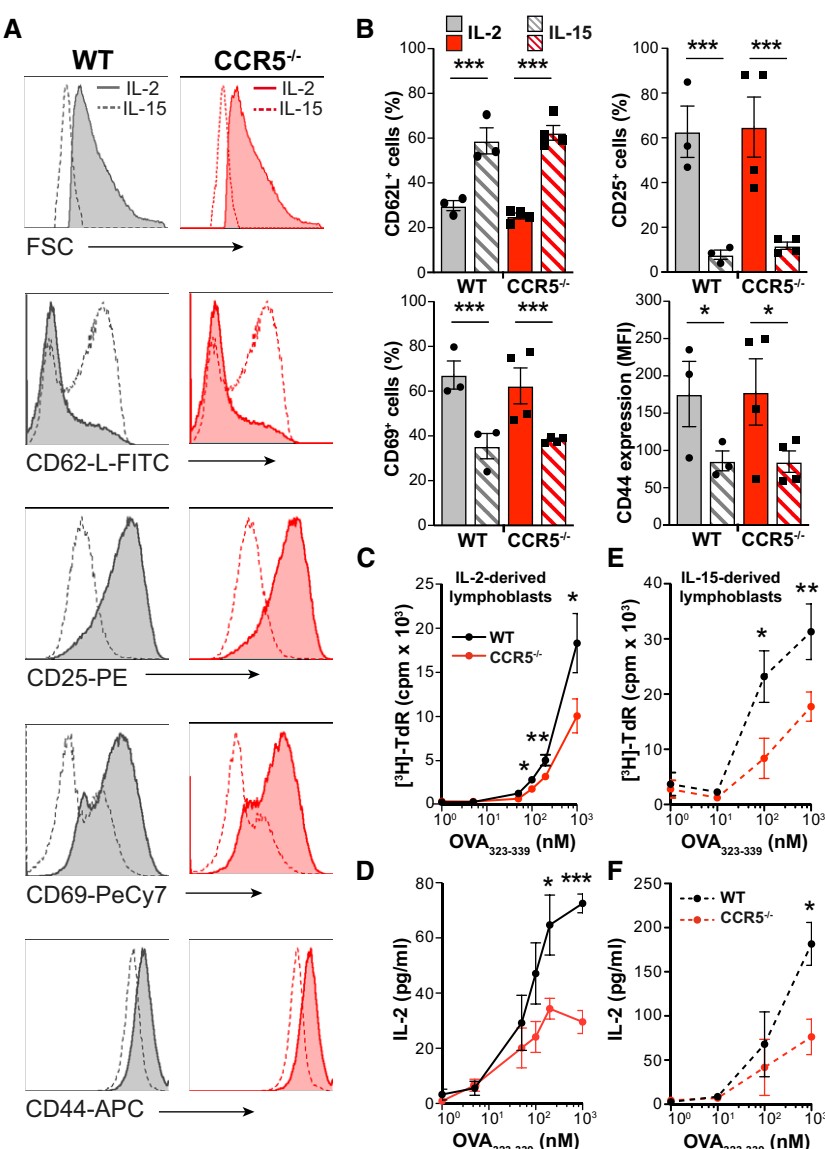

**Figure 2. CCR5 increases the sensitivity of antigen-experienced CD4$^+$ T cells.**

A, B   Representative histograms and quantification of mean fluorescence intensity (MFI; A) or the percentage of cells positive for the indicated memory markers (B) in OT-II WT and CCR5$^{-/-}$ lymphoblasts expanded in IL-2 or IL-15, as specified. Data shown as mean ± SEM (*n* ≥ 3). The gating strategy is shown in Fig EV1.

C–F   IL-2- (C, D) and IL-15-expanded lymphoblasts (E, F) were restimulated with indicated concentrations of OVA$_{323-339}$; cell proliferation (thymidine incorporation into DNA; C, E) and IL-2 production (by ELISA; D, F) were measured after 72 h. Data are presented as mean ± SEM (*n* = 5).

Data information: \*$P < 0.05$, \*\*$P < 0.01$, \*\*\*$P < 0.001$, two-way ANOVA (B) or two-tailed unpaired Student's *t*-test (C–F).

nanoclusters between OT-II WT and CCR5$^{-/-}$ naïve cells, which had a small percentage of TCR nanoclusters larger than 4 TCR in both genotypes (Fig 3A). In contrast, there was a significant increase in TCR nanocluster number and size in WT compared to CCR5$^{-/-}$ lymphoblasts (Fig 3B and C). The number of TCR nanoclusters per cell analyzed in each condition is also indicated (Appendix Table S1). As predicted, there was a gradient in TCR nanoclustering of naïve ≪ IL-2- < IL-15-differentiated OT-II WT cells (Appendix Fig S1E), which coincided with increased antigenic sensitivity of the IL-15-expanded cells (Appendix Fig S1F and G). These findings thus reinforce the IL-15-induced memory-like phenotype versus the IL-2-induced effector-like phenotype and link TCR nanoclustering with increased sensitivity in antigen-experienced CD4$^+$ T cells. The difference in TCR nanoclustering between WT and CCR5$^{-/-}$ cells was nevertheless similar in IL-2- and IL-15-expanded lymphoblasts, which indicates that CCR5 affects TCR nanoclustering in lymphoblasts independently of the cytokine milieu.

Using a Monte Carlo simulation, we applied data from surface replicas of naïve and IL-2-expanded OT-II lymphoblasts to determine whether the experimental frequency of cluster size was due to random distribution of gold particles. In all cases, the cluster distributions observed experimentally differed significantly from pure random proximity between clusters (Appendix Fig S2). To define the differences between OT-II WT and CCR5$^{-/-}$ cells, we used a model that accounts for receptor clustering dynamics (Castro *et al*, 2014), a Bayesian inference method that estimates the so-called clustering parameter, *b*. Based on this model, we concluded that the probability of a chance nanocluster distribution similar to that observed for naïve and activated OT-II WT and CCR5$^{-/-}$ cells approaches 0% (Fig 3D and E). Posterior distribution analysis also showed that whereas the clustering parameter was very similar between naïve OT-II WT and CCR5$^{-/-}$ cells (Fig 3D), there was clear separation in lymphoblasts (Fig 3E). These analyses provide a mathematical framework that validates the TCR nanoclustering differences between WT and CCR5$^{-/-}$ cells, as determined by EM.

The differences in TCR oligomerization between OT-II WT and CCR5$^{-/-}$ lymphoblasts were also studied using blue-native gel electrophoresis (BN-PAGE) (Schamel *et al*, 2005; Swamy & Schamel, 2009). Cell lysis with digitonin, a detergent that disrupts TCR nanoclusters into their monomeric components, showed that WT and CCR5$^{-/-}$ lymphoblasts expressed comparable TCR levels, as detected with anti-CD3ζ antibodies (Fig 3F). Cell lysis with Brij96, which preserves TCR nanoclusters, showed a notable reduction in large TCR complexes in CCR5$^{-/-}$ compared to WT lymphoblasts (Fig 3F). Two independent techniques thus support a CCR5 role in TCR nanoscopic organization in antigen-experienced CD4$^+$ T cells.

To determine whether CCR5 controls TCR nanoclustering in *in vivo*-generated memory T cells, we analyzed TCR distribution in surface replicas of CD4$^+$ memory T cells purified by negative selection from OVA/OVA-immunized WT and CCR5$^{-/-}$ mice (Appendix Fig S3). CD4$^+$ memory cells from CCR5$^{-/-}$ mice showed fewer, smaller TCR nanoclusters than those from WT counterparts (Fig 3G; Appendix Table S1), which indicates that CCR5 promotes formation of large TCR nanoclusters in endogenously generated CD4$^+$ memory T cells.

## CCR5-induced TCR nanoclustering is independent of its co-stimulatory activity

Since CCR5 has co-stimulatory functions in CD4$^+$ T-cell priming (Molon *et al*, 2005; González-Martín *et al*, 2011), it is of interest to know whether defective TCR clustering in CCR5$^{-/-}$ lymphoblasts is due to suboptimal primary activation of these cells. To address this question, we treated OT-II WT cells with the CCR5 antagonist TAK-779 at various intervals throughout culture and analyzed TCR nanoclusters in IL-2-expanded T lymphoblasts. TAK-779 addition during the priming phase (blockade of CCR5 co-stimulatory function) decreased the percentage of large TCR nanoclusters compared to untreated controls (Fig 4A). TAK-779 treatment did not alter TCR clustering in OT-II CCR5$^{-/-}$ cells (Appendix Fig S4), which indicates that the TAK-779 effect on OT-II cells is CCR5-specific.

To avoid interference with the CCR5 co-stimulatory activity, we primed OT-II WT cells in the absence of the inhibitor and added TAK-779 only during IL-2-driven expansion of the CD4$^+$ lymphoblasts. In these conditions, TAK-779 also reduced the percentage of large TCR nanoclusters (Fig 4B), which indicates that the CCR5 signals that control TCR organization are independent of those involved in its co-stimulatory function.

We next explored whether other chemokine receptors involved in T-cell activation control TCR nanoclusters in CD4$^+$ T cells. CXCR4 is a paradigmatic chemokine receptor that also provides co-stimulatory signals (Kumar *et al*, 2006; Smith *et al*, 2013). We primed OT-II WT cells in the presence of the CXCR4 antagonist AMD3100 and analyzed TCR nanoclusters in IL-2-expanded T lymphoblasts. Vehicle- and AMD3100-treated cells showed similar TCR nanocluster distribution (Fig 4D), which implies that CXCR4 blockade does not interfere with TCR nanoclustering.

## CCR5 deficiency increases ceramide levels in CD4$^+$ T cells

We analyzed CCR5 regulation of TCR nanoclustering in CD4$^+$ T cells and found no differences between OT-II WT and CCR5$^{-/-}$ cells in TCR/CD3 chain mRNA levels or in cell surface expression of the TCRα chain (Fig EV3). These data suggest that the reduction in TCR clustering in CCR5$^{-/-}$ cells is not due to decreased TCR expression.

T-cell antigen receptor nanoclustering is dependent on plasma membrane Chol and SM (Molnar *et al*, 2012), two lipids also necessary for CCR5 signaling (Mañes *et al*, 2001). OT-II WT and CCR5$^{-/-}$ lymphoblasts expressed comparable levels of total Chol and SM species (Fig 5A and B). OT-II CCR5$^{-/-}$ lymphoblasts nonetheless showed a significant increase in most ceramide (Cer) species and their dihydroCer (dhCer) precursors (Fig 5C and D). These differences were not observed in naïve OT-II WT and CCR5$^{-/-}$ cells (Appendix Fig S5A), indicative that the Cer increase was specific to antigen-experienced cells. The increase in Cer species in CCR5$^{-/-}$ lymphoblasts was not linked to enhanced apoptosis compared to WT cells (Appendix Fig S5B).

## CCR5 deficiency upregulates specific ceramide synthases in CD4$^+$ T cells

Our analysis of the mRNA levels of key enzymes involved in Cer metabolism showed no differences in ceramidases (*ASAH1, ACER 2,*

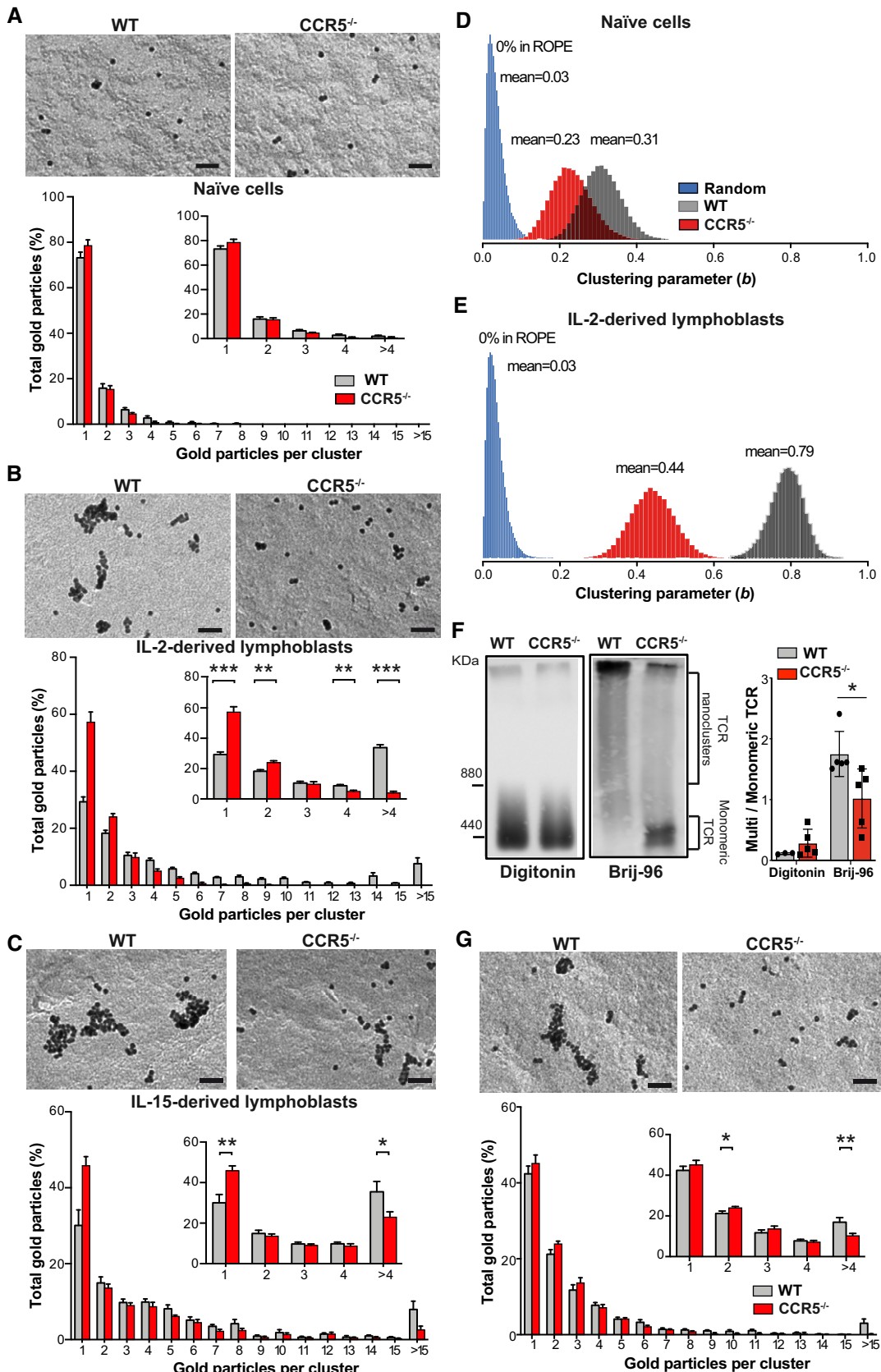

Figure 3.

**Figure 3.  CCR5 increases TCR nanoclustering in antigen-experienced CD4$^+$ T cells.**

A–C   Analysis of TCR nanoclustering by EM in OT-II WT and CCR5$^{-/-}$ naïve cells (A; $n$ = 6 cells/genotype; WT: 3,427, CCR5$^{-/-}$: 3,528 particles), and IL-2- (B; WT, $n$ = 8 cells, 15,419 particles; CCR5$^{-/-}$, $n$ = 6 cells, 5,410 particles) or IL-15-expanded lymphoblasts (C; WT, $n$ = 8 cells, 27,518 particles; CCR5$^{-/-}$, $n$ = 7 cells, 22,696 particles). A representative small field image at the top of each panel shows gold particle distribution in the cell surface replicas of anti-CD3ε-labeled cells; at bottom, quantification (mean ± SEM) of gold particles in clusters of indicated size in WT (gray bars) and CCR5$^{-/-}$ cells (red). Insets show the distribution of clusters of one, two, three, four, or more than four particles, and statistical analysis.

D, E   Posterior distribution in naïve (D) and IL-2-expanded lymphoblasts (E) of the clustering parameter $b$ for WT (gray) and CCR5$^{-/-}$ cells (red); randomly generated distributions of receptors are shown in blue. The mean value of the $b$ parameter is indicated for each condition. The probability of a chance distribution similar to that determined in cells is nearly 0% by the ROPE.

F   Comparison of TCR oligomer size using BN-PAGE and anti-CD3ζ immunoblotting in day 10, IL-2-expanded WT and CCR5$^{-/-}$ OT-II lymphoblasts lysed in buffer containing digitonin or Brij-96. The marker protein is ferritin (f1, 440 and f2, 880 kDa forms). The ratio of TCR nanoclusters to monomeric TCR in each lysis condition was quantified by densitometry (right; $n$ = 5).

G   Top, representative small field EM images showing gold particle distribution in the cell surface replicas of CD4$^+$ T cells isolated from OVA/OVA-immunized WT and CCR5$^{-/-}$ mice. Bottom, quantification (mean ± SEM) of gold particles in clusters of the indicated size (WT, gray bars; $n$ = 5 cells, 14,680 particles; CCR5$^{-/-}$, red; $n$ = 7 cells, 15,374 particles). Insets show the distribution between clusters of one, two, three, four, or more than four particles, and statistical analysis.

Data information: (A–C, F, G), Data are mean ± SEM. *$P$ < 0.05, **$P$ < 0.01, ***$P$ < 0.001, one-tailed unpaired Student's $t$-test. Scale bar, 50 nm (A–C, G).

---

ACER 3) and sphingomyelinases (*SMPD1–4*) between OT-II WT and CCR5$^{-/-}$ naïve cells or lymphoblasts (Fig EV4). mRNA levels of the ceramide synthases (CerS) CerS2, CerS3, and CerS4 were nonetheless upregulated in OT-II CCR5$^{-/-}$ lymphoblasts (Fig 5E); CerS5 and CerS6 were unaltered, and the nervous system-specific CerS1 isoenzyme was not detected. CerS2, CerS3, and CerS4 levels were comparable in naïve CD4$^+$ WT or CCR5$^{-/-}$ cells (Fig 5E), which again associate the CCR5 transcriptional effect on these genes with activation.

We sought to validate the CerS isoforms upregulated by CCR5 deficiency at the protein level. In accordance with mRNA analyses, CerS2 protein levels were significantly higher in CCR5$^{-/-}$ than in WT lymphoblasts (Fig 5F); CerS3 and CerS4 were undetectable or only barely detectable by immunoblot. This is consistent with the fact that CerS2 has the highest expression level and the broadest substrate specificity in other cell types (Laviad *et al*, 2008). Chromatin immunoprecipitation (ChIP), followed by amplification of a region of the CerS2 promoter enriched in CpG islands, showed that binding of the transcriptional activation marker acetylated histone H3K9 (H3K9Ac) was higher in CCR5$^{-/-}$ than in WT lymphoblasts (Fig 5G). Moreover, blockade of CCR5 signaling with pertussis toxin (PTx; an inhibitor of the Gα$_i$ subunit) also increased CerS2 mRNA expression (Fig 5H).

To further study CCR5 transcriptional regulation of CerS, we scanned for transcription factors with putative binding sites in the CerS2, CerS3, and CerS4 promoters, which are transcriptionally upregulated in CCR5$^{-/-}$ lymphoblasts, but not represented in the CerS6 promoter, which is not CCR5-regulated. We selected two regions; region 1 comprised −5 kb to the 5′UTR, and region 2 encompassed the 5′UTR to the first coding exon (Fig 5I). This bioinformatic approach identified GATA-1 and NF-IB (nuclear factor-1B) as putative transcription factors involved in the differential expression of the CerS2 isoform (Fig 5J and K).

We focused on GATA-1, since it is implicated in the differentiation of some CD4$^+$ T-cell subtypes (Sundrud *et al*, 2005; Fu *et al*, 2012). Immunofluorescence analyses showed increased nuclear levels of the phosphoSer142-GATA-1 form in OT-II CCR5$^{-/-}$ compared to WT lymphoblasts (Fig 5L and M), which correlated with enriched GATA-1 binding to the CerS2 promoter in CD4$^+$ CCR5$^{-/-}$ lymphoblasts (Fig 5N). CCR5 deficiency might induce CerS2 transcription through GATA-1.

## Ceramide levels control TCR nanoclustering

We used a synthetic biology approach to determine whether ceramide content affects TCR nanoclustering. Large unilamellar vesicles (LUV) were prepared at different molar ratios of PC, Chol, SM, and Cer (Fig 6A) and then reconstituted with a streptavidin-binding-peptide-tagged TCR purified in its native state from murine M.mζ-SBP (streptavidin-binding peptide) T cells (Swamy & Schamel, 2009). The proteoliposomes were analyzed by BN-PAGE after solubilization in 0.5% Brij96 to maintain TCR nanocluster integrity or in 1% digitonin to disrupt TCR clusters. As anticipated (Molnar *et al*, 2012; Wang *et al*, 2016), TCR was monomeric in PC-containing LUV, whereas it formed nanoclusters when reconstituted in PC/Chol/SM liposomes (Fig 6B and C). The inclusion of ceramides in these LUV (PC/Chol/SM/Cer liposomes) reduced TCR nanoclustering in a dose-dependent manner. This effect was not due to differential TCR reconstitution in Cer-containing LUV, since digitonin treatment rendered equivalent levels of monomeric TCR in each condition (Fig 6B). These data suggest that Cer membrane content impairs TCR nanoclustering.

To test whether this effect also occurs in live cells, we treated OT-II WT lymphoblasts with recombinant sphingomyelinase (SMase), which hydrolyzes SM to ceramide (Kitatani *et al*, 2008). SMase treatment of WT OT-II blasts increased Cer levels robustly (Fig 6D), but did not compromise cell viability (Appendix Fig S6). Analysis of membrane replicas from these cells showed that SMase treatment reduced the number of high valency TCR nanoclusters compared to controls (Fig 6E; Appendix Table S1), which indicates that high Cer levels hinder TCR nanoclustering in CD4$^+$ T cells.

## CerS2 silencing restores TCR nanoclustering after CCR5 functional blockade

To correlate increased CerS2 expression with the impaired TCR nanoclustering in OT-II CCR5$^{-/-}$ T cells, we attempted to silence CerS2 expression by lentiviral transduction of primary lymphoblasts with short-hairpin (sh) RNA for CerS2 or control (shCtrl). In the most successful experiments, we were only able to transduce ~ 20% of the lymphoblasts, which did not lead to solid CerS2 mRNA silencing (Appendix Fig S7A and B). Despite the low efficiency, antigenic restimulation tended to promote stronger responses in shCerS2-

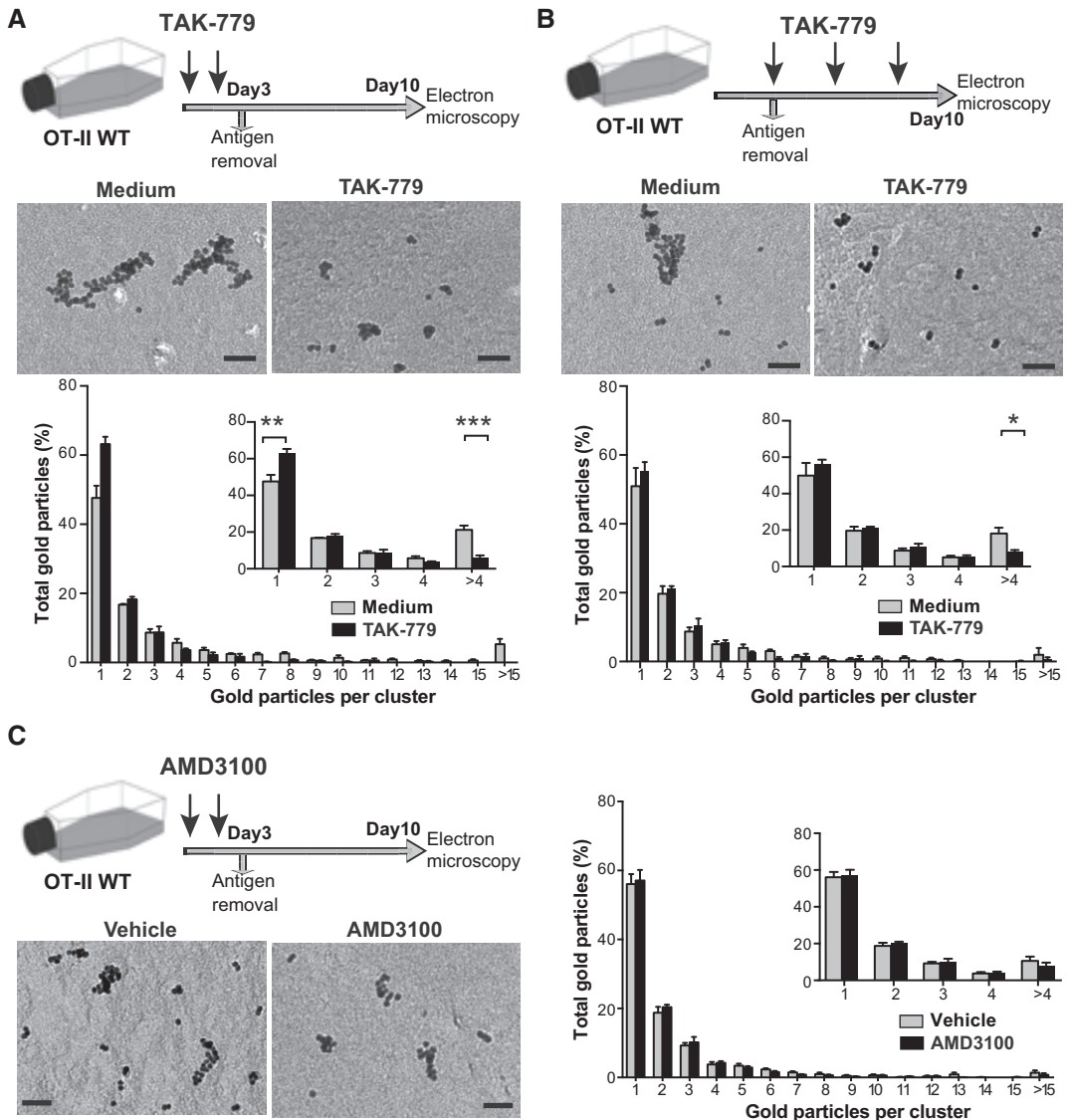

**Figure 4. CCR5-induced TCR nanoclustering is specific and independent of its co-stimulatory activity.**

A  OT-II WT cells were activated with OVA$_{323-339}$, alone or with TAK-779. After 3 days, antigen and TAK-779 were removed and lymphoblasts expanded in IL-2-containing medium. TCR nanoclustering was analyzed in anti-CD3ε-labeled surface replicas of day 10 lymphoblasts. Top, representative small field EM images showing gold particle distribution in the cell surface replicas of WT CD4$^+$ T cells alone or with TAK-779. Bottom, quantification of gold particles in clusters of the indicated size. Inset, distribution of gold particles between clusters of one, two, three, four, or more than four particles in vehicle- (gray bars; n = 5 cells, 11,266 particles) and TAK-779-treated cells (black; n = 6 cells, 5,138 particles).

B  OT-II WT cells were activated with OVA$_{323-339}$, and TAK-779 was added at days 3, 5, and 7 after antigen removal. Analysis as above, untreated (gray bars; n = 5 cells, 6,400 particles) and TAK-779-treated cells (black; n = 6 cells, 7,153 particles). Inset shows the distribution between clusters of one, two, three, four, or more than four particles, and statistical analysis.

C  OT-II WT naïve cells were activated with antigen in the presence or not of the CXCR4 inhibitor AMD3100. Left, representative EM images showing gold particle distribution in the cell surface replicas. Right, analysis of gold particles in clusters as above, vehicle- (gray bars; n = 6 cells, 12,339 particles) and AMD3100-treated cells (black; n = 7 cells, 17,059 particles). Inset shows the distribution between clusters of one, two, three, four, or more than four particles, and statistical analysis.

Data information: (A–C), Data are mean ± SEM. *P < 0.05, **P < 0.01, ***P < 0.001, one-tailed unpaired Student's t-test. Scale bar, 50 nm.

than in shCtrl-transduced cells (Appendix Fig S7C and D). The low efficiency also precluded analysis of TCR nanoclusters in membrane replicas, as transduced cells could not be distinguished from non-transduced cells.

To overcome these difficulties, we used the 2B4 CD4$^+$ T-cell line. We verified that 2B4 cells expressed CCR5 and that TAK-779 treatment increased CerS2 levels and impaired TCR nanoclustering (Appendix Fig S8). The data suggest that CCR5 effects on TCR nanoclustering and CerS2 induction are not exclusive to the OT-II system and that TAK-779-treated 2B4 cells mimic the functional findings in OT-II CCR5$^{-/-}$ lymphoblasts. 2B4 cells were transduced efficiently by lentiviruses and, after

3 days of antibiotic selection, 100% of the cells expressed the shRNA; this led to strong silencing of CerS2 mRNA and protein in shCerS2- compared to shCtrl-transduced cells (Fig 6F–H). Analysis of TCR organization showed recovery of large TCR nanoclustering in TAK-779-treated, shCerS2-transduced cells compared to controls (Fig 6I; Appendix Table S1); after restimulation with plate-bound anti-CD3ε antibody in the presence of

TAK-779, CD69 upregulation was higher in CerS2-deficient than in shCtrl-cells (Fig 6J).

### CCR5 modulates TCR nanoclustering in human CD4+ T cells

Finally, we tested whether CCR5 deficiency also impairs TCR organization in human CD4+ T cells. Approximately 1% of the

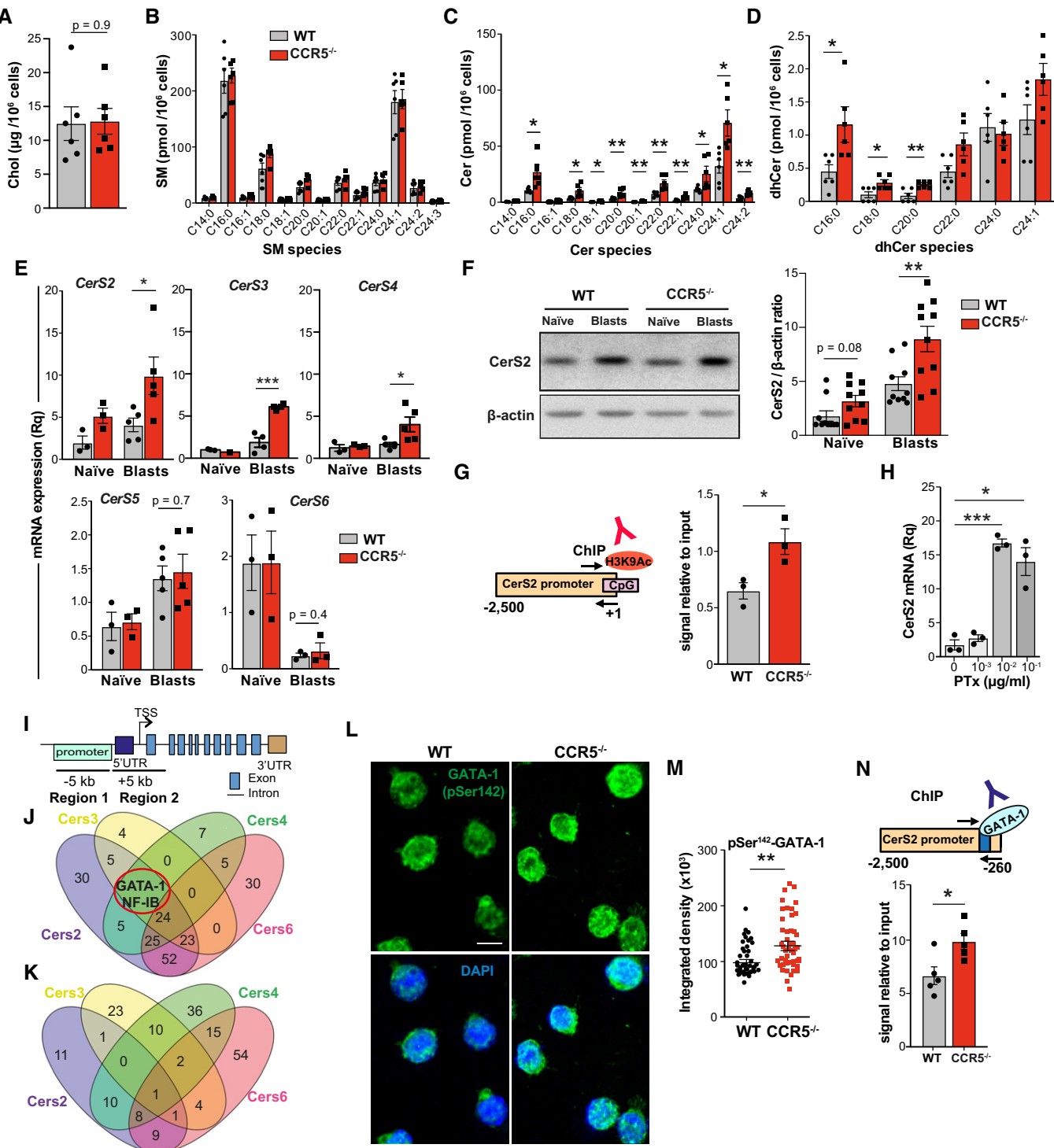

**Figure 5.**

◀

**Figure 5. CCR5 deficiency increases Cer levels by upregulating specific CerS.**

A   Total Chol levels in WT and CCR5$^{-/-}$ OT-II lymphoblasts (day 10, IL-2-expanded) as determined by a fluorometric assay ($n = 6$).

B–D   SM (B), Cer (C) and dhCer (D) levels in WT and CCR5$^{-/-}$ OT-II 10-day lymphoblasts, as determined by UPLC-TOF MS. Values, after normalization with C17 standards and cell number in each sample, are the mean of two independent experiments ($n = 6$).

E   RT–qPCR determination of CerS mRNA levels in naïve and IL-2-expanded WT and CCR5$^{-/-}$ OT-II 10-day lymphoblasts ($n = 3$–5).

F   Representative immunoblot showing CerS2 protein levels in naïve and WT and CCR5$^{-/-}$ OT-II 10-day lymphoblasts, and densitometric quantification of blots as above ($n = 10$).

G   ChIP analysis of the CerS2 promoter using an anti-H3K9Ac antibody. Scheme of the CerS2 promoter showing CpG islands and primers used for amplification. Relative ChIP of the CerS2 promoter in WT and CCR5$^{-/-}$ OT-II 10-day lymphoblasts ($n = 3$).

H   Relative CerS2 mRNA level in CD4 T cells treated with PTx ($n = 3$).

I   Scheme of a canonical CerS gene to illustrate the *in silico* strategy used to search for CerS-specific transcription factors.

J, K   Venn diagrams showing the number of transcription factors with putative binding sites in the indicated CerS genes in regions 1 (J) and 2 (K). The red circle highlights the transcription factors shared by CerS2, CerS3, and CerS4 promoters, but not present in the CerS6 promoter.

L   Representative immunofluorescence images showing pSer142-GATA-1 staining (green) of OT-II WT and CCR5$^{-/-}$ lymphoblasts. The green channel (top) and the merge with nuclear DAPI staining (blue; bottom) are shown. Scale bar, 10 μm.

M   Quantification of nuclear staining of the cells plotted as integrated density fluorescence intensity in DAPI-stained area ($n \geq 50$ cells/condition).

N   Top, basic scheme of the CerS2 promoter, indicating the putative GATA-1 binding site (blue) and location of the primers used for amplification in ChIP assays (black arrows). Bottom, relative anti-GATA-1 ChIP levels in OT-II WT and CCR5$^{-/-}$ lymphoblasts ($n = 5$).

Data information: (E, G, H, N), Each data point is the average of triplicates in an independent experiment. (A–H, M, N), Data shown as mean $\pm$ SEM of triplicates; *$P < 0.05$, **$P < 0.01$, ***$P < 0.001$, two-tailed unpaired Student's $t$-test.

Source data are available online for this figure.

Spanish population bears the *ccr5Δ32* polymorphism in homozygosity (Mañes *et al*, 2003). Purified CD4$^+$ T cells from healthy WT or *ccr5Δ32* homozygous donors were activated with anti-CD3 and anti-CD28 antibodies for 3 days and maintained for five additional days with IL-2. We found that *ccr5Δ32* lymphoblasts had a lower percentage of large TCR nanoclusters than WT cells (Fig 7A; Appendix Table S1); concomitantly, the fraction of monomeric TCR was increased in the former. Sphingolipid analysis of these CD4$^+$ lymphoblasts showed an increase in saturated 24-carbon Cer (C24:0) and its precursor (dhCer C24:0) in cells derived from *ccr5Δ32* donors, whereas SM levels were comparable between both genotypes (Fig 7B). This increase in Cer levels was associated with upregulation of CerS2 mRNA in *ccr5Δ32* lymphoblasts compared to WT controls (Fig 7C); expression of other enzymes involved in Cer metabolism was unchanged in both genotypes (Fig EV5). These results indicate that, as found in mouse CCR5$^{-/-}$ lymphoblasts, antigen-experienced human CD4$^+$ T cells from *ccr5Δ32* homozygotes show defective TCR nanoclustering associated with increased Cer levels and upregulated CerS2. Moreover, they indicate that these CCR5 effects are not restricted to specific T-cell clones, but can be observed in a polyclonal T-cell repertoire.

## Discussion

Here, we show that CCR5 signaling is largely dispensable for memory CD4$^+$ T-cell differentiation, but provides specific signals that improve the functional fitness of memory cells after antigen re-encounter. The CCR5 signals optimize TCR nanoclustering and antigen sensitivity by triggering a CD4$^+$ T-cell-specific transcription program that regulates Cer metabolism. This CCR5 program operates in murine and human CD4$^+$ T cells, which suggests physiopathological relevance.

A central observation of our study is that CCR5 expression enhances the degree of TCR nanoclustering in resting antigen-experienced T cells, both *in vitro* and *in vivo*. The presence of TCR nanoclusters in resting T cells was shown by BN-PAGE, EM, and super-resolution microscopy (Schamel *et al*, 2005; Hu *et al*, 2016; Jung *et al*, 2016; Pageon *et al*, 2016). We demonstrate here differential TCR nanoclustering in WT and CCR5-deficient cells using two complementary approaches (EM and BN-PAGE), based on different conceptual principles. In EM, TCR nanoclusters were defined as gold particle aggregates at < 10 nm distance from one another. Previous analyses showed that this criterion permits identification of TCR nanoclusters formed by TCR-TCR interactions (Kumar *et al*, 2011); these tightly associated TCR nanoclusters would allow inter-TCR cooperativity for pMHC binding (Martín-Blanco *et al*, 2018). We therefore intentionally considered TCR not to be in the same nanocluster if the gap between them was > 10 nm; this excludes considering more loosely associated TCR as nanoclusters, but the strict definition allowed association of TCR nanoclusters to a T-cell biological function.

It is also important to clarify that gold particle counts do not necessarily correspond to the number of TCR molecules in a nanocluster. BN-PAGE defines neither the exact size nor the abundance of TCR nanoclusters. Direct comparison of CCR5$^{-/-}$ with WT cells using both methods nonetheless allowed us to detect relative differences and determine the promoter effect of CCR5 in TCR nanoclustering. Application of Monte Carlo simulations further indicated that the nanoclusters observed in EM are not the result of random proximity of gold particles. The estimated clustering parameter ($b$) for randomly distributed particles was virtually zero.

Since CCR5 provides positive signals during activation of naïve CD4$^+$ T cells (Molon *et al*, 2005; Nesbeth *et al*, 2010; González-Martín *et al*, 2011), we attempted to clarify whether TCR nanoclustering impairment in CCR5$^{-/-}$ cells is solely an effect of this defective priming. This is unlikely, since TCR clustering was inhibited when CCR5 was blocked during expansion of fully activated WT lymphoblasts. This effect on TCR nanoclustering during lymphoblast expansion was modest compared to that observed in the priming phase, but is probably the result of insufficient CCR5 inhibition during lymphoblast expansion. CCR5 is not only upregulated shortly after activation, but is maintained in memory CD4$^+$ T cells, which are highly susceptible to

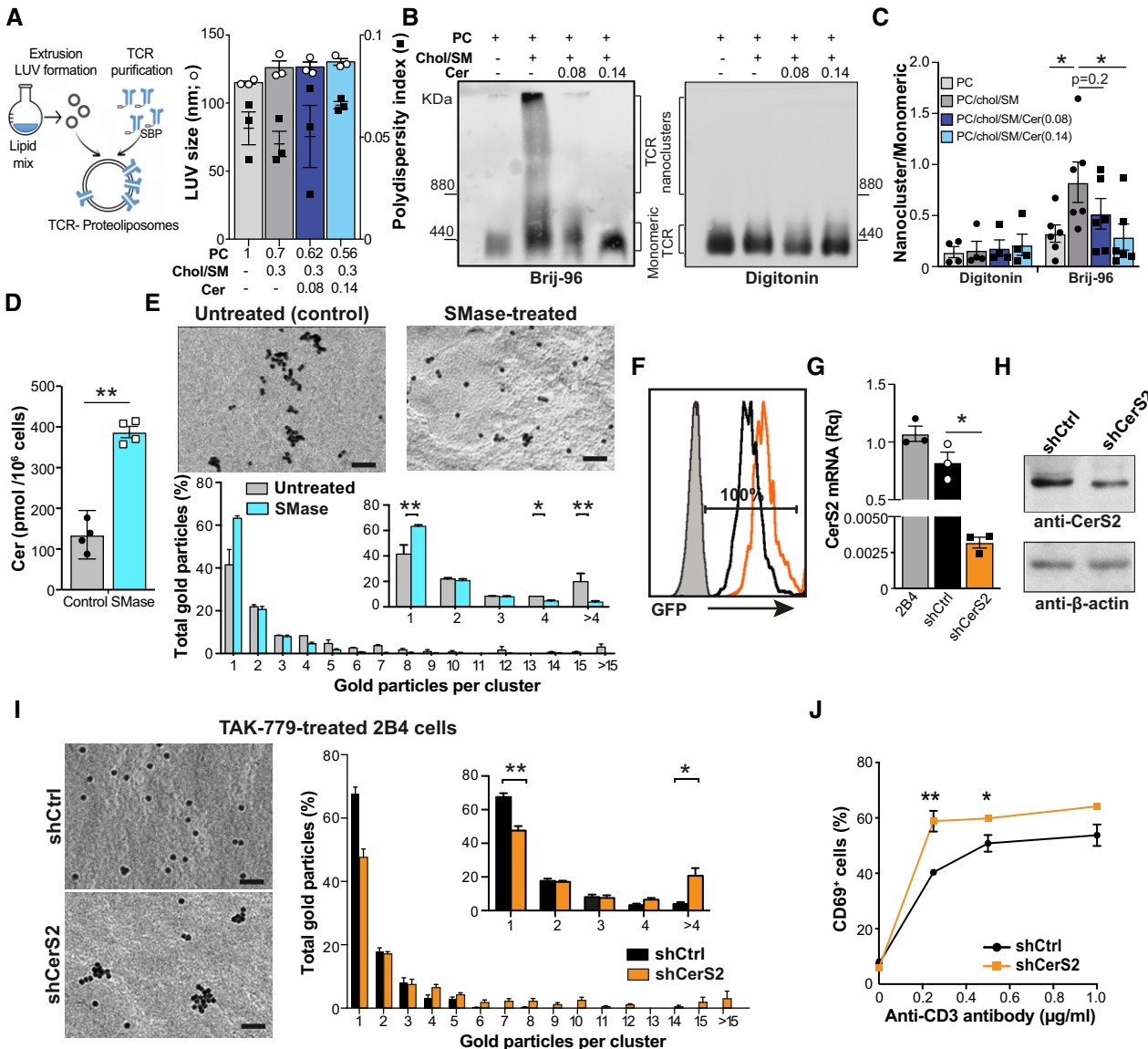

**Figure 6. Cer levels determine the grade of TCR nanoclustering.**

A   Scheme of the strategy used to form TCR proteoliposomes, and size of LUV generated at the indicated lipid molar ratio. Polydispersity index values are shown as black squares for each condition (n = 3). SBP, streptavidin-binding-peptide-tagged TCR.

B   Representative immunoblots comparing TCR nanocluster sizes via BN-PAGE and anti-CD3ζ immunoblotting in TCR proteoliposomes lysed in the presence of Brij-96 or digitonin. The marker protein is ferritin (f1, 440 and f2, 880 kDa forms).

C   The ratio of the nanocluster and monomeric TCR in each lysis condition was quantified by densitometry from immunoblots as in (B) (n ≥ 4).

D   Cer levels in OT-II 10-day lymphoblasts, untreated or treated with SMase (n = 4).

E   Representative small field images showing gold particle distribution, and quantification (mean ± SEM) of gold particles in clusters of the indicated size in cell surface replicas from untreated (gray bars; n = 5 cells, 8,126 particles) and SMase-treated (1 h) OT-II lymphoblasts (cyan; n = 6 cells, 8,457 particles) after CD3ε labeling, as determined by EM. The inset shows distribution between clusters of one, two, three, four, or more than four particles.

F   GFP expression in shCtrl- (black) and shCerS2 (orange)-transduced 2B4 cells after puromycin selection, as determined by FACS. Non-transfected 2B4 cells (gray).

G   Relative CerS2 mRNA levels in TAK-779-treated 2B4 cells as in (F). Values were normalized to those obtained in untransduced TAK-779-treated 2B4 cells (n = 3).

H   Representative immunoblot with anti-CerS2 antibody to determine CerS2 protein levels in shCtrl and ShCerS2-transduced 2B4 cells as in (G). Filters were rehybridized with β-actin as loading control.

I   TCR nanoclustering of shCtrl- and shCerS2-transduced 2B4 cells in the presence of TAK-779 as determined by EM. Representative small field images and quantification (mean ± SEM) of gold particles in clusters of indicated sizes in cell surface replicas shCtrl (black bars; n = 6 cells, 12,337 particles) and shCerS2 2B4 lymphoblasts (orange; n = 7 cells, 13,456 particles). Inset, distribution between clusters of indicated size and statistical analysis.

J   Percentage of CD69+ shCtrl (black) and shCerS2 (orange) 2B4 cells restimulated with plate-bound anti-CD3ε antibody in the presence of TAK-779 (n = 3).

Data information: (A, C–E, G–J), Data are shown as mean ± SEM. *P < 0.05, **P < 0.01, one-tailed (E, I) or two-tailed unpaired Student's t-test. Scale bar, 50 nm.
Source data are available online for this figure.

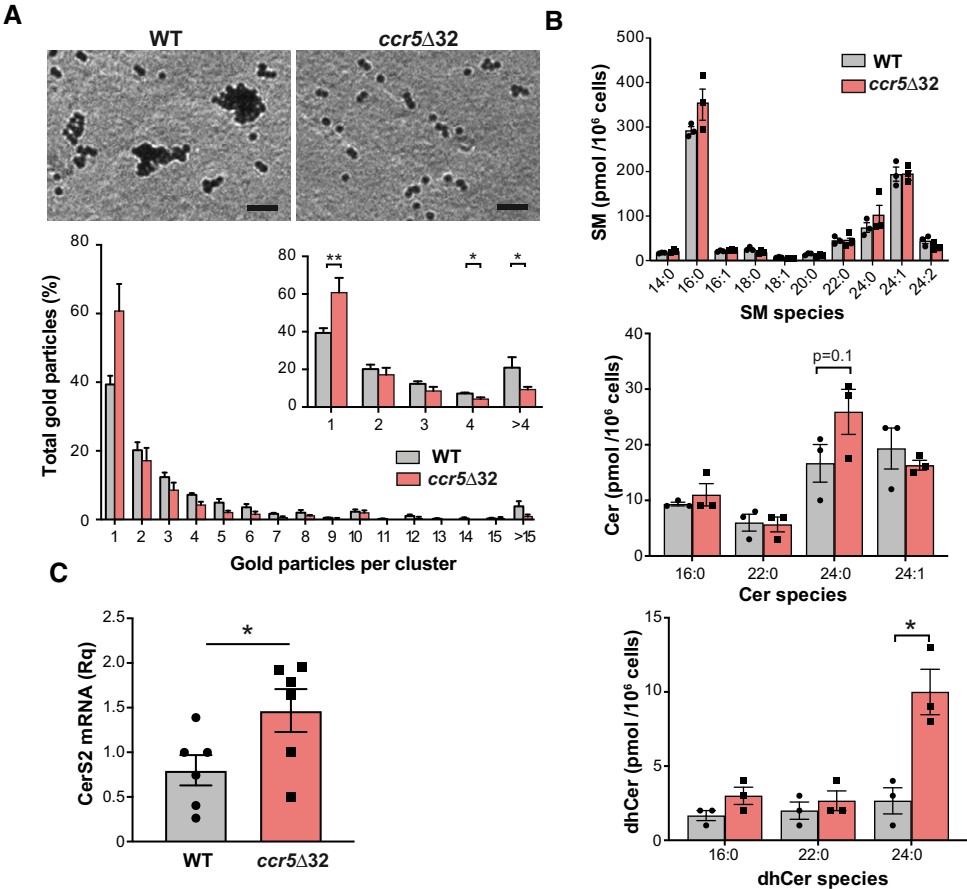

**Figure 7. CCR5 controls TCR nanoclustering and Cer metabolism in human CD4⁺ cells.**

A  Analysis of TCR nanoclustering in lymphoblasts from healthy WT and *ccr5*Δ32 homozygous donors by EM. Top, representative small field image showing gold particle distribution in cell surface replicas of anti-CD3ε-labeled cells; bottom, quantification (mean ± SEM) of gold particles in clusters of the indicated size in the WT (gray bars; *n* = 5 cells, 17,689 particles) and Δ32/Δ32 cells (light red; *n* = 4 cells, 16,938 particles). Insets show the distribution between clusters of one, two, three, four, or more than four particles, and statistical analysis.

B  Normalized SM, Cer, and dhCer levels in lymphoblasts obtained as in (A). A representative experiment is shown (*n* = 3 donors/genotype; *n* = 2 independent experiments).

C  Relative CerS2 mRNA levels in day 8 WT and *ccr5*Δ32 lymphoblasts. Each data point is the average of a technical triplicate from three donors in two independent experiments (*n* = 6).

Data information: Data are shown as mean ± SEM (B, C). *$P < 0.05$, **$P < 0.01$, two-tailed unpaired Student's *t*-test. Scale bar, 50 nm.

infection by R5-tropic HIV-1 strains (Nie *et al*, 2009). Our results showed increased CCR5 mRNA expression during lymphoblast expansion. These lymphoblasts also expressed CCR5 ligands, suggesting autocrine CCR5 stimulation during this phase. We thus propose that TCR nanoclustering is regulated by CCR5 signals transduced during lymphoblast differentiation rather than during priming.

Another feature that distinguishes CCR5 effects on priming and on TCR nanoclustering is the role of CXCR4 in these events. During priming, CCR5 and CXCR4 are recruited to and accumulate as heteromeric complexes at the IS of CD4⁺ T cells; AMD3100 (a CXCR4 antagonist) prevented not only CXCR4 but also CCR5 accumulation (Contento *et al*, 2008), which indicates necessary cooperation between CCR5 and CXCR4 for full T-cell activation. In contrast, AMD3100 did not affect TCR nanoclusters in CCR5-expessing cells, which suggests that CXCR4/CCR5 heterodimer signaling is not

essential for TCR nanoclustering in lymphoblasts. CCR5 homo- and heterodimers are thought to associate differently with Gα subunits; homodimers signal through the PTx-sensitive Gαᵢ, whereas heterodimers generate PTx-resistant responses (Mellado *et al*, 2001). CCR5 co-stimulatory signals in the IS are PTx-resistant (Molon *et al*, 2005), consistent with CCR5/CXCR4 heterodimerization during priming. PTx potentiates transactivation of the CerS2 promoter (Fig 5H), however, which suggests involvement of the CCR5-induced Gαᵢ pathway in TCR nanoclustering. CCR5 thus appears to trigger distinct signaling pathways for co-stimulation and TCR nanoclustering in CD4⁺ T cells. Since chemokine receptors can form nanoclusters (Martinez-Muñoz *et al*, 2018), it would be of interest to study potential feedback loops between TCR and CCR5 nanoclusters.

Cholesterol and SM are two lipids essential for CCR5 signaling and TCR nanoclustering (Mañes *et al*, 2001; Molnar *et al*, 2012). In

resting T cells, these receptors nonetheless partition in different membrane phases, liquid-ordered ($l_o$) for CCR5 (Molon *et al*, 2005) and liquid-disordered ($l_d$) for TCR (Beck-Garcia *et al*, 2015). This differential phase segregation argues against direct CCR5/TCR interaction as a mechanism that influences TCR nanoclustering. Our results suggest instead that increased levels of long-chain Cer species cause defective TCR nanoclustering in CCR5$^{-/-}$ lymphoblasts. Indeed, elevation of Cer levels in TCR-reconstituted proteoliposomes and in live cells by SM hydrolysis impaired nanoscopic TCR organization. Although Cer levels were higher in lymphoblasts than in naïve cells, which supports a role for Cer in T-cell activation (Sofi *et al*, 2017), CCR5 deficiency further increased Cer levels specifically in lymphoblasts. The Cer increase in CCR5$^{-/-}$ lymphoblasts did not cause spontaneous apoptosis (Appendix Fig S6B), which coincides with the non-apoptotic and preventive effects of long-chain Cer in this process (Stiban & Perera, 2015).

Although Cer levels increased in antigen-experienced CD4$^+$ T cells from CCR5$^{-/-}$ mice and ccr5Δ32 homozygotes, there were differences between the mouse and the human cell ceramidome. This could depend on many factors, including the overall species-specific enzymes involved in sphingolipid metabolism, or the lipid composition of the diet. Membrane phase segregation properties of ceramides are dependent not only on the saturation, but also on the length of their acyl chain and the lipid microenvironment that surrounds them (Alonso & Goñi, 2018). Although C24:1 is the most abundant Cer in mouse WT and CCR5$^{-/-}$ T cells, the biggest differences associated with CCR5 deficiency were observed for C20:0 and C22:0 Cer. The differential membrane phase segregation properties of C24:0 and C24:1 Cer might be subtle in an environment enriched in other long-chain saturated Cer and dhCer.

Several settings can be hypothesized that explain Cer effects on TCR nanoclustering. In model membranes, Cer has strong segregation capacity, which might affect their lateral organization (Alonso & Goñi, 2018). When the Chol concentration is saturating, however, as is the case of cell membranes and our proteoliposomes, Cer-enriched domains are not formed, due to the ability of Chol and Cer to displace one another (Castro *et al*, 2009). This argues against the idea that Cer impairs TCR clustering by promoting a general reduction in membrane lateral diffusion. Our mathematical model also predicted that cluster distribution is independent of TCR diffusivity. Interaction of the TCRβ subunit with Chol/SM complexes is critical for TCR nanoclustering (Molnar *et al*, 2012; Wang *et al*, 2016). High levels of Cer or their precursors (dhCer) could transform Chol/SM into Chol/SM/Cer domains with specific physicochemical properties, which might hinder TCR nanocluster formation physically or thermodynamically. For instance, high dhSM levels inhibit CCR5-mediated HIV-1 infection by rigidifying CCR5-containing $l_o$ domains (Vieira *et al*, 2010). Chol/SM interaction depends on the hydrogen bond generated by the amide group of the SM molecule and the 3-hydroxyl group of Chol (Ramstedt & Slotte, 2002), but the SM amide group can also form hydrogen bonds with the Cer hydroxyl group (Garcia-Arribas *et al*, 2016). It is thus possible that, rather than forming SM/Chol/Cer domains, small increases in Cer levels would increase the mutual displacement of three lipids. This could lead to replacement of SM/Chol by SM/Cer complexes with gel-like biophysical properties (Sot *et al*, 2008).

Our results indicate that the CCR5 effect on TCR clustering is biologically meaningful. In a first model, we show that the responses of CCR5-deficient memory CD4$^+$ T cells generated by vaccination were impaired after *ex vivo* stimulation. In a second model that involves T:B-cell cooperation, we show that CCR5 deficiency impaired class switching of high-affinity antibodies after re-exposure to a T cell-dependent antigen. Affinity maturation and class switching depend on recruitment of $T_{fh}$ cells to GC (Vinuesa *et al*, 2016). This $T_{fh}$ cell confinement is a result of CXCR5 expression and the downregulation of other homing receptors, including CCR5 (Crotty, 2011), which could explain the lack of difference in class switching between OVA/KLH-immunized WT and CCR5$^{-/-}$ mice. There were also no differences between WT and CCR5$^{-/-}$ effector $T_{fh}$ cells after the first OVA immunization. Once GC resolve, however, some $T_{fh}$ cells are reported to enter the circulation as $T_{fh}$ central memory-like cells (Vinuesa *et al*, 2016). These circulating, antigen-experienced $T_{fh}$ cells express CCR5 and are very susceptible to HIV-1 infection (Xu *et al*, 2017). We found that the frequency of large TCR nanoclusters increased in memory T cells from WT compared to CCR5$^{-/-}$ OVA/OVA-immunized mice, which suggests increased antigenic sensitivity.

We hypothesize that following re-exposure to antigen, CCR5-expressing memory pre-$T_{fh}$ cells will have a more efficient response than CCR5-deficient cells, which would support robust antibody responses after their differentiation to GC-$T_{fh}$ cells. In humans, functional CCR5 deficiency does not cause strong immune suppression, but ccr5Δ32 homozygosity was associated with four times more fatal infections than average during the 2009–2011 influenza season in Spain (Falcon *et al*, 2015) and fatal infections by the West Nile Virus in the United States (Lim & Murphy, 2011). Our results provide a conceptual framework on which to base clinical trials to evaluate CD4$^+$ T-cell memory responses in CCR5-deficient humans, and suggest caution regarding the risks associated with genetic ablation of CCR5 as a preventive strategy to block HIV-1 infection.

## Material and Methods

Resource Identification Portal accession numbers for antibodies, cell lines, animals, and other reagents used in the study are provided in Appendix Table S2.

### Antibodies and reagents

Antibodies used to characterize mouse cells by flow cytometry were anti-Vα$_2$TCR-PE (B20.1), anti-CD25-PE (PC61), anti-CD45.2-FITC (104), anti-CD62L-FITC/APC (MEL-14), anti-CD69-PeCy7 (H1.2F3), and biotinylated anti-CXCR5 (2G8) from BD Biosciences; anti-human biotinylated CD3 (OKT3), anti-CD4-PeCy7/eFluor450/Pacific Blue (RM4.5), anti-IFNγ-APC (XMG1.2), and anti-PD1-eFluor780 (J43) from eBioscience; and anti-CD44-PeCy5/APC (IM7) from BioLegend. Biotinylated and purified anti-CD3ε (145-2C11; BD Biosciences) were used for EM and T-cell activation, respectively. Anti-mouse CerS-2 (1A6; Novus Biologicals), anti-CD3ζ (449, purified from hybridoma), and anti-β-actin (AC-15; Sigma-Aldrich) were used for immunoblot. Anti-mouse GATA-1 (D52H6; Cell Signaling) and anti-mouse phospho-GATA1$^{pSer142}$ (Thermo Fisher Scientific) were used for immunofluorescence. Anti-GATA1 (ab11852, Abcam), anti-histone H3Lys9 (CS200583), and purified IgG rabbit (PP64B; EMD Millipore) were used for ChIP.

The OVA$_{323-339}$ peptide was synthesized at the CNB Proteomics facility. TAK-779, AMD-3100, poly-L-lysine, pertussis toxin, Cer (bovine spinal cord), and sphingomyelinase (*Bacillus cereus*) were from Sigma-Aldrich. mCCL4, mIL-2, and mIL-15 were from Pepro-Tech; NIP-KLH, NIP-OVA, NIP(7)-BSA, and NIP(41)-BSA were from Biosearch Technology. Soybean phosphatidylcholine Chol, egg SM, C12 Cer (d18:1/12:0), C16 Cer (d18:1/16:0), C18 Cer (d18:1/18:0), C24 Cer (d18:1/24:0), C24:1 Cer (d18:1/24:1(15Z)), C16 dhCer (d18:0/16:0), C18 dhCer (d18:0/18:0), C24 dhCer (d18:0/24:0), C24:1 dhCer (d18:0/24:1(15Z)), C12:0 SM (d18:1/12:0), C16:0 SM (d18:1/16:0), C18:0 SM (d18:1/18:0), C24:0 SM, C24:1 SM, and the Cer mix from bovine spinal cord were from Avanti Polar Lipids. Lentiviral pGIPZ containing shRNAs for murine Cers 2 (V3LMM_454307, V3LMM_454309, and V3LMM_454311 clones), and the mismatched control were from Dharmacon.

## Mice and cell lines

C57BL/6J WT and CCR5$^{-/-}$ mice were from The Jackson Laboratory. TCR transgenic OT-II CCR5$^{-/-}$ mice, recognizing OVA$_{323-339}$ (ISQAVHAAHAEINEAGR; I-Ab MHC class II molecule), have been described (González-Martín *et al*, 2011). B6-SJL (Ptprca Pepcb/ BoyJ) mice bearing the pan-leukocyte marker allele CD45.1 were used for adoptive transfer experiments. CD3ε-deficient mice (DeJarnette *et al*, 1998) were used as a source of antigen-presenting cells for restimulation assays. Mice were maintained in SPF conditions in the CNB and CBM animal facilities, in accordance with national and EU guidelines. All animal procedures were approved by the CNB and the Comunidad de Madrid ethical committees (PROEX 277/14; PROEX 090/19). Human embryonic kidney HEK-293T cells and the murine 2B4 hybridoma and its derivative M.mζ-SBP (which expresses a SBP-tagged form of CD3ζ) (Swamy & Schamel, 2009) were cultured in standard conditions.

## Isolation and culture of mouse and human primary T cells

Spleen and lymph nodes from 6- to 12-week-old OT-II WT and CCR5$^{-/-}$ mice were isolated and cell suspensions obtained using 40-μM pore filters. Erythrocytes were lysed with AKT lysis buffer (0.15 M NH$_4$Cl, 10 mM KHCO$_3$, 0.1 mM EDTA), and cells were activated with the appropriate OVA peptide for 3 days. Antigen was removed, and cells were cultured with IL-2 (5 ng/ml) or IL-15 (20 ng/ml). For some experiments, naïve OT-I and OT-II cells were obtained by negative selection using the Dynabeads Untouched Mouse CD4 Cell Kit (Thermo Fisher). Flow cytometry indicated > 85% enrichment in all cases. Memory CD4$^+$ T cells, generated *in vivo* after NIP-OVA or NIP-KLH immunization (see below), were isolated by negative selection with the Mouse Memory T cell CD4$^+$/ CD62L$^-$/CD44$^{hi}$ Column Kit (R&D Systems).

Blood samples from *ccr5Δ32* homozygous and WT healthy donors were from the Fundació ACE (Barcelona, Spain) and obtained with informed consent of the donors. No personal data were registered, and all procedures using these samples were in accordance with the standards approved by the Ethics Committee of the Hospital Clinic Barcelona (HCB/2014/0494 and HCB/2016/ 0659). Human peripheral blood mononuclear cells (PBMC) were isolated from Vacutainer Cell Preparation Tubes by separation on a Ficoll gradient. CD4$^+$ T cells were obtained by negative selection

using the EasySep Human CD4$^+$ Enrichment kit (Stem Cell Technologies) and stimulated with anti-CD3-coated magnetic beads (Dynabeads M-450 tosyl-activated, Thermo Fisher) for 3 days. Beads were removed with a magnet, and cells were incubated with IL-2 (5 ng/ml) to generate lymphoblasts. The *ccr5Δ32* polymorphism (rs333) was genotyped by PCR (AriaMx Real-time; Agilent Technologies) as described (Mañes *et al*, 2003).

## Flow cytometry

For cell surface markers, cell suspensions were incubated (20 min, 4°C) with the indicated fluorochrome-labeled or biotinylated monoclonal primary antibodies in phosphate-buffered saline with 1% BSA and 0.02% NaN$_3$ (PBS staining buffer). For intracellular labeling, cells were fixed and permeabilized with IntraPrep (Beckman Coulter), followed by intracellular staining with indicated antibodies. Cells were analyzed on Cytomics FC500 or Gallios cytometers (both from Beckman Coulter) and data analyzed using FlowJo software.

## Immunization and adoptive transfer

Spleen and lymph node cell suspensions from OT-II WT or CCR5$^{-/-}$ cells were adoptively transferred (5 × 10$^6$ cells/mouse) into CD45.1 mice. The following day, recipient mice were infected intravenously with rVACV-OVA virus (2 × 10$^6$ pfu). Mice were sacrificed 35 days later, and splenocyte suspensions were obtained as described above.

C57BL/6J or CCR5$^{-/-}$ mice were immunized (i.p.) with NIP-OVA (200 μg) in alum (100 μl) diluted 1:1 in PBS. At 7 days postimmunization, spleens from three mice of each genotype were harvested and analyzed by flow cytometry to detect T$_{fh}$ cells. On day 30, mice were randomized in a blind manner and half of the mice in each group were re-immunized with NIP-OVA/alum (as above); the other half received NIP-KLH (200 μg)/alum. Mice were sacrificed 15 days later, and serum anti-NIP antibodies were determined by ELISA. Plate-bound NP(7)-BSA and NP(41)-BSA (5 μg/ ml) were used to measure high- and low-affinity Ig, respectively. Sera from NIP-OVA- and NIP-KLH-immunized mice were diluted 1:175, and, after several washing steps, anti-NIP antibody binding was developed with the SBA Clonotyping System-HRP (Southern Biotech). Absorbance at 405 nm was determined in a FilterMax F5 microplate reader (Molecular Devices). Memory cells from NIP-OVA- and NIP-KLH-immunized mice were purified as indicated and processed for EM.

## Immunogold labeling, replica preparation, and EM analysis

Immunogold-labeled cell surface replicas were obtained as described (Kumar *et al*, 2011). Briefly, T cells were fixed in 1% paraformaldehyde (PFA) and labeled with anti-mouse CD3 mAb (145-2C11) or anti-human CD3 mAb OKT3, followed by 10 nm gold-conjugated protein A (Sigma-Aldrich). Labeled cells were adhered to poly-L-lysine-coated mica strips and fixed with 0.1% glutaraldehyde. Samples were covered with another mica strip, frozen in liquid ethane (KF-80, Leica), and stored in liquid nitrogen. Cell replicas were prepared with a Balzers 400T freeze fracture (FF) unit, mounted on copper grids, and analyzed on a

JEM1010 electron microscope (Jeol, Japan) operating at 80 kV. Images were taken with a CCD camera (Bioscan, Gatan, Pleasanton, CA) and processed with TVIPS software (TVIPS, Gauting, DE). EM images were collected by two researchers, one of them blind to the experiment. Gold particles were counted on the computer. When distance between gold particles was smaller than their diameter (10 nm), they were considered part of the same cluster.

### BN-PAGE analysis of TCR clustering

Membrane fractions from OT-II WT and CCR5$^{-/-}$ cells ($20 \times 10^6$) were prepared with a Dounce homogenizer, followed by incubation in hypotonic buffer (10 mM HEPES pH 7.4, 42 mM KCl, 5 mM MgCl$_2$, protease inhibitors). Membranes were recovered by ultracentrifugation (100,000 $g$, 45 min, 4°C) and lysed in 150 μl BN lysis buffer (500 mM Bis-Tris 40 mM pH 7.0, 1 mM ε-aminocaproic acid, 40 mM NaCl, 20% glycerol, 4 mM EDTA, and 0.5% Brij96 or 1% digitonin) with protease inhibitors. BN-PAGE gradient gels (4–8%) were prepared and used as described (Swamy & Schamel, 2009), using ferritin 24-mer and 48-mer (f1, 440 kDa; f2, 880 kDa) as protein markers. Proteins were transferred to PVDF membranes and probed with anti-CD3ζ antibody.

### Restimulation assays

Splenocytes from CD3ε$^{-/-}$ mice were irradiated (15 Gy), seeded ($0.6 \times 10^5$ cells/well), and loaded (2 h, 37°C) with different concentrations of OVA$_{323–339}$ peptide. After centrifugation (300 $g$, 5 min), isolated lymphoblasts ($0.75 \times 10^5$ cells/well) were co-cultured for 48 h. Supernatants were collected to measure IL-2 by ELISA (ELISA MAX Deluxe, BioLegend) and proliferation was assessed by methyl-$^3$[H]-thymidine (1 μCi/well) incorporation into DNA, in a 1450 Microbeta liquid scintillation counter (Perkin-Elmer).

### TCR purification

The TCR fused to streptavidin-binding peptide was purified from M.mζ-SBP cells. Briefly, $100 \times 10^6$ cells were lysed in lysis buffer (20 mM Bis-Tris pH 7, 500 mM ζ-aminocaproic acid, 20 mM NaCl, 2 mM EDTA, 10% glycerol, 1% digitonin). After incubation of the lysate with streptavidin-conjugated agarose (overnight, 4°C), the TCR was eluted by incubating samples with 2 mM biotin in lysis buffer (30 min, 4°C).

### Preparation of large unilamellar vesicles and TCR reconstitution

Large unilamellar vesicles with a custom lipid composition were prepared by the thin film method (Molnar et al, 2012), followed by extrusion through polycarbonate membranes with a pore size of 200 nm (21 times) and 80 nm (51 times). The diameter of the resulting LUV was determined by dynamic light scattering (Zeta-master S, Malvern Instruments). The LUV preparation (2 mM) was mixed with purified TCR (0.1 μg) in 100 μl saline-phosphate buffer with 0.02% Triton X-100, and 40 μl 0.01% Triton X-100 was added. Samples were agitated (30 min, 4°C), and detergent was removed by adsorption to polystyrene Bio-Beads SM-2 (3 mg;

Bio-Rad; overnight, 4°C). Proteoliposomes were collected by ultracentrifugation (180,000 $g$, 4 h, 4°C), lysed, and analyzed by BN-PAGE as above.

### Sphingolipid and Chol quantification

Total Chol level was measured with the Amplex Red Cholesterol Assay Kit (Invitrogen) after lysis (50 mM Tris–HCl pH 8; 150 mM NaCl, 1% NP-40). Sphingolipid determinations were performed by an external researcher blind to the experimental groups. Prior to sphingolipid quantification, calibration curves were prepared with mixtures of C12Cer, C16Cer, C18Cer, C24Cer, C24:1 Cer, C16dhCer, C18dhCer, C24dhCer, C24:1dhCer, C12SM, C16SM, C18SM, C24SM, and C24:1SM. For sphingolipid determination, cell pellets ($1 \times 10^6$) were mixed with internal standards (*N*-dode-canoyl-sphingosine, *N*-dodecanoylglucosylsphingosine, *N*-dodeca-noyl-sphingosylphosphorylcholine, C17-sphinganine, and C17-sphinganine-1 phosphate; 0.2 nmol each; Avanti Polar Lipids) in a methanol:chloroform solution. Sphingolipids were extracted as described (Merrill et al, 2005), solubilized in methanol, and analyzed by ultra-performance liquid chromatography (UPLC; Waters, Milford, MA) connected to a time-of-flight detector (TOF; LCT Premier XE) controlled by Waters/Micromass MassLynx software. Lipid species were identified based on accurate mass measurement with an error < 5 ppm, and their LC retention time was compared with the standard ($\pm$ 2) (Muñoz-Olaya et al, 2008).

### SMase treatment

All experiments (sphingolipid quantification, apoptosis, and TCR nanoclustering) were performed by incubating OT-II WT and CCR5$^{-/-}$ cells ($0.2 \times 10^6$) with recombinant sphingomyelinase from *Bacillus cereus* (0.5 U/ml; 1 h, 37°C) in serum-free medium. Cells were washed and processed immediately for EM analysis or for sphingolipid quantification as above.

### Quantitative RT–PCR analyses

Total RNA was extracted from human or murine cells using the RNeasy Mini Kit (Qiagen), and cDNA was synthesized from 1 μg total RNA (High Capacity cDNA Reverse Transcription Kit, Promega). Quantitative RT–PCR was performed using FluoCycle II SYBR Master Mix (EuroClone) with specific primers (Appendix Table S3) in an ABI 7300 Real-Time PCR System (Applied Biosystems). Results were analyzed using SDS2.4 software.

### CerS2 silencing

Lentiviruses were produced in HEK-293T cells after co-transfection with pGIPZ-shRNA-CerS2 or control plasmids, pSPAX2 and pMD2.G (VSV-G protein) using LipoD293tm (SignaGen). Supernatants were concentrated by ultracentrifugation and supplemented with poly-brene (8 μg/ml). Lymphoblasts (3 days post-activation) or 2B4 cells ($1.5 \times 10^6$ cells/ml) were resuspended in lentiviral supernatant and centrifuged (900 $g$, 90 min, 37°C). Transduction efficiency was analyzed after 24 h by FACS. In the case of 2B4 cells, transduced

cells were selected with puromycin (2 μg/ml) for 3 days prior to analyses.

## Immunofluorescence analyses

OT-II 10-day WT or CCR5$^{-/-}$ lymphoblasts were plated in poly-L-lysine-coated coverslips (Nunc Lab-Tek Chamber Slide, Thermo Scientific; 50 μg/ml, overnight, 4°C). After adhesion (1 h, 37°C), cells were fixed in 4% PFA (10 min), Triton-X100-permeabilized (0.3% in PBS, 15 min), and blocked with BSA 0.5% in PBS. Samples were incubated (overnight, 4°C) with anti-mouse phospho-GATA1$^{pSer142}$ antibody (1/200), followed by anti-rabbit Ig Alexa-488 secondary antibody (1 h). Coverslips were mounted in Fluoro-mount-G with DAPI (Southern Biotech); images were acquired with a Zeiss LSM710 and analyzed by a blind observer with NIH ImageJ software.

## ChIP assay

Chromatin immunoprecipitation assays were performed with the EZ-ChIP Kit (Millipore). In brief, OT-II WT or CCR5$^{-/-}$ lymphoblasts ($2 \times 10^7$) were fixed (1% PFA, 10 min, RT) and quenched (125 mM glycine, 5 min, RT). Cells were harvested ($1 \times 10^7$ cells/ml), lysed (15 min, 4°C), and DNA sheared by sonication (45 cycles; 30 s on/off; Bioruptor Pico, Diagenode) in aliquots (0.2 ml). Of each lysate, 1% was stored as input reference, and the remaining material was immunoprecipitated (14 h, 4°C, with rotation) with antibodies to GATA1, histone H3-Lys9, or purified IgG (control). Immune complexes were captured using Protein G Magnetic Beads (Bio-Rad) and, after washing, eluted with 100 mM NaHCO$_3$, 1% SDS; protein/DNA bonds were disrupted with proteinase K (10 μg/μl, 2 h, 62°C). DNA was purified using spin columns, and Cers2 gene promoter sequences were analyzed with specific primers (Appendix Table S3). The relative quantity of amplified product in the input and ChIP samples was calculated (Mira *et al*, 2018).

## Cell migration and Ca²⁺ flux assays

OT-II WT or CCR5$^{-/-}$ lymphoblasts ($10^6$) were added to the upper chamber of a Transwell (3-μm pore; Corning) and allowed to migrate toward 100 nM CCL4 for 4 h. Migrating cells were quantified by flow cytometry (Cytomics FC500). Mobilization of intracellular Ca$^{2+}$ stores after CCL4 (100 nM) stimulation was measured as reported (Gómez-Moutón *et al*, 2015).

## Mathematical and Bayesian analyses

To analyze cluster size distribution, we used a standard chi-square test to compare the fraction of clusters of a given size (1, 2, 3, etc.) in each dataset. In all plots, "Random" refers to synthetic distributions of receptors generated randomly.

To quantify the mechanistic relevance of cluster size between random distributions of clusters and clusters in WT and in CCR5$^{-/-}$ CD4$^+$ T cells, we used a Bayesian inference model on top of a mechanistic model (Castro *et al*, 2014). The model assumes that TCR aggregates by incorporating one receptor at a time, with on and off rates that depend on the diffusion properties of the receptor on the membrane, but not existing cluster size. That is,

$$1 \underset{q-}{\overset{q-}{\rightleftharpoons}} 2 \underset{q+}{\overset{q-}{\rightleftharpoons}} 3 \underset{q+}{\overset{q-}{\rightleftharpoons}} \ldots \underset{q+}{\overset{q-}{\rightleftharpoons}} n-1 \underset{q+}{\overset{q-}{\rightleftharpoons}} n \underset{q+}{\overset{q-}{\rightleftharpoons}} n+1\ldots$$

The "affinity" of the process is given by $b = q+/q-$, which we also refer to as the clustering or affinity parameter. In the steady state, we can calculate analytically the fraction of clusters of a given size $n$:

$$\pi_n = \frac{b^{n-1}(1-b)}{(1-b^{N_{max}})}$$

with

$$b < 1, \quad n \in \{1, 2, 3, \ldots, N_{max}\}.$$

The model was fitted using the Bayesian JAGS code (Kruschke, 2014) (see Appendix Supplementary Methods). The histograms for the number of clusters of a given size $n$ ($N_n$) were modeled as a multinomial distribution with the number of observations, $N$, given by the total count per experiment, and probabilities $\pi_n$ given by the formulas above. The priors for the clustering parameter $b$ are beta distributions with shape parameters $A$ and $B$ with non-informative uniform priors. Specifically,

$N_n \sim$ Multinomial $(\pi_{n, N})$
$b \sim$ Beta $(A, B)$
$A \sim$ Uniform $(0, 1,000)$
$B \sim$ Uniform $(0, 1,000)$

Posterior distribution of the estimated clustering parameter, $b$, is given with the so-called Region of Practical Equivalent (ROPE), defined as the probability of a parameter from a dataset to be explained by another dataset. ROPE quantifies the probability that the observed clustering parameter (and distribution of clusters) in the experiment can be obtained by pure random proximity.

At the molecular level, the kinetic rates $q^+$ and $q^-$ can be expressed in terms of the diffusion rates of the receptors, $k_d^{+/-}$, the receptor size ($a$), the mean distance between receptors ($s$), and the correct receptor–receptor binding rates, $k^{+/-}$, through the equations (Lauffenburger & DeLisi, 1983):

$$q+ = \frac{k_d^+ k^+}{k_d^+ + k^-} \quad q- = \frac{k_d^- k^-}{k_d^+ + k^-} \Rightarrow b = \frac{k_d^+ k^+}{k_d^- + k^-}$$

with

$$k_d^+ = \frac{4\pi D}{\log(s/a) - 3/4} \quad k_d^- = \frac{2\pi D}{s^2(\log(s/a) - 3/4)}.$$

The clustering parameter $b$ is independent of the TCR diffusivity (as $D$ is canceled), so the observed TCR nanoclustering differences for WT and CCR5$^{-/-}$ cells would be due to TCR-TCR interactions, as previously reported (Beck-Garcia *et al*, 2015).

### Identification of transcription factors in CerS promoters

Cers2, Cers3, Cers4, and Cers6 gene coordinates were obtained from the UCSC Genome browser (https://genome.ucsc.edu/; mouse genome version GRCm38/mm10). Known transcription factors for these genes were identified at GTRD v17.04 (http://gtrd17-04.b iouml.org/). Venn diagrams were constructed to identify common and specific transcription factors for ceramide synthase genes.

### Statistical analyses

For comparison between two conditions, data were analyzed using parametric Student's *t*-tests, paired when different treatments were applied to the same sample, or unpaired with Welch's correction. Multiple parametric comparisons were analyzed with one-way ANOVA with Bonferroni's *post-hoc* test. The chi-square test was used to analyze overall distribution of gold particles. F test was used to compare variances. All analyses were performed using Prism 6.0 or 7.0 software (GraphPad). Differences were considered significant when $P < 0.05$.

## Data and code availability

This study includes no data deposited in external repositories. The authors confirm that all relevant data and materials supporting the findings of this study are available on reasonable request. This excludes materials obtained from other researchers, who must provide their consent for transfer. The Bayesian JAGS code generated in the study is provided as supplementary information in the Appendix Supplementary Methods.

**Expanded View** for this article is available online.

## Acknowledgements
We thank D Sancho and JW Yewdell for rVACV-OVA virus, RM Peregil for technical assistance, MC Moreno and S Escudero for flow cytometry services (CNB), MT Rejas and M Guerra for EM service (CBM-SO), and C Mark for excellent editorial assistance. Fundació ACE would like to thank patients, controls, and the staff who participated in this project. This work was funded by grants from the Spanish Ministerio de Ciencia, Innovación y Universidades (SAF2017–83732-R to SM; FIS2016-78883-C2-2-P to MC; CTQ2017-85378-R; AEI/FEDER, EU), the Instituto de Salud Carlos III (ISCIII) (PI13/02434, PI16/01861 to AR), the Comunidad de Madrid (B2017/BMD-3733; IMMUNOTHERCAN-CM to SM), and the Merck-Salud Foundation (to SM). WWS and CD were supported by the Deutsche Forschungsgemeinschaft (DFG) through BIOSS-EXC294 and CIBSS-EXC 2189 (Project 390939984), SCHA976/7-1, and SFB1381. The Genome Research @ Fundació ACE project (GR@ACE) is supported by the Fundación Bancaria La Caixa, Grifols SA, Fundació ACE, and ISCIII (Ministry of Health, Spain). Fundació ACE is a participating center in the Dementia Genetics Spanish Consortium (DEGESCO).

## Author contributions
SM conceived the study. AM-L and RB designed, performed most experiments, and interpreted data. JC and GF performed lipid analyses, and CD performed liposome experiments. ER-B performed analysis of TCR chains. MC carried out mathematical simulations and Bayesian analyses. MES carried out bioinformatic analyses. IR, LMR, and AR performed *ccr5Δ32* genotyping and provided samples. WWAS, HMS, and BA contributed ideas and technical support. SM and RB wrote the manuscript. All authors read, discussed, and edited the manuscript.

## Conflict of interest
The authors declare that they have no conflict of interest.

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
