## [Review Process File · The EMBO Journal]

CCR5 deficiency impairs CD4+ T cell memory responses and antigenic sensitivity through increased ceramide synthesis

Ana Martín-Leal, Raquel Blanco, Josefina Casas, María Sáez, Elena Bovolenta, Itziar de Rojas, Carina Drechsler, Luis Miguel Real, Gemma Fabriás, Agustín Ruiz Laza, Mario Castro, Wolfgang Schamel, Balbino Alarcon, Hisse van Santen, and Santos Mañes

DOI: [10.15252/embj.2020104749](https://doi.org/10.15252/embj.2020104749)

Corresponding author(s): Santos Mañes (smanes@cnb.csic.es)

Review Timeline:

Submission Date:	18th Feb 20
Editorial Decision:	23rd Mar 20
Revision Received:	18th Apr 20
Editorial Decision:	11th May 20
Revision Received:	12th May 20
Accepted:	14th May 20

Editor: Karin Dumstrei

Transaction Report:

Dear Prof. Mañes,

Thank you for submitting your manuscript for consideration by the EMBO Journal. It has now been seen by three referees whose comments are shown below.

Thank you for submitting your manuscript to The EMBO Journal. Your study has now been seen by two referees and their comments are provided below. As you can see below, the referees appreciate that the analysis adds new insight and find the study the interesting. They raise a number of good points that I would like to invite you to address in a revised version. Regarding point Ref #2 point #4 I don't know if you have any genetic data to address this point or if there is a way to do this experiment using inhibitors. I am happy to discuss the raised points further and maybe it would be most helpful to do so via phone or skype. I will contact you in the few days to discuss this further.

I should add that it is EMBO Journal policy to allow only a single major round of revision, and that it is therefore important to resolve the major concerns at this stage. acceptance of your manuscript will therefore depend on the completeness of your responses in this revised version.

Thank you for the opportunity to consider your work for publication. I look forward to your revision.

Yours sincerely,

Karin Dumstrei, PhD
Senior Editor
The EMBO Journal

When assembling figures, please refer to our figure preparation guideline in order to ensure proper formatting and readability in print as well as on screen:
<http://bit.ly/EMBOPressFigurePreparationGuideline>

Further information is available in our Guide For Authors:

The revision must be submitted online within 90 days; please click on the link below to submit the revision online before 21st Jun 2020.

Link Not Available

Referee #1:

In the present work, Martin-Leal et al. characterized the role of CCR5, a coreceptor for HIV infection and an important chemokine receptor regulating trafficking of effector and memory T cells in inflamed tissues, on the sensitivity of "antigen-experienced" CD4⁺ memory T cells to cognate antigens. To do that, they analysed *in vivo* CD4⁺ T cell memory responses by transferring lymph node/spleen cell suspensions from OT-II CCR5 WT or OT-II CCR5^{-/-} mice to congenic CD45.1 mice and infecting them with OVA-encoding vaccinia virus. They found that CCR5 deficiency did not affect the percentage of effector/memory CD4⁺ T cells or T follicular helper cells but impaired memory T cell responses to antigenic re-stimulation in terms of reduced percentage of IFN- γ producing cells and reduced generation of high affinity class-switched Abs in B cells. They also demonstrated that the *in vivo* defective memory responses induced by CCR5 deficiency were associated to a reduced antigen sensitivity of antigen-experienced CD4⁺ T cells. In order to understand the molecular basis of the reduced antigen sensitivity of antigen-experienced CD4⁺ T cells in the absence of CCR5, they evaluated variations in number and size of TCR nanoclusters at the surface of both naïve cells and lymphoblasts from WT and CCR5^{-/-} mice by using electron microscopy analyses and blue-native gel electrophoresis. Both independent techniques evidenced that in the absence of CCR5, TCR nanoclusters on lymphoblasts were smaller in both number and size than WT. They also demonstrated that the reduction of TCR nanoclustering in CCR5^{-/-} CD4⁺ T cells was associated to an increase of ceramide and its precursor dihydroceramide due to transcriptional up-regulation of ceramide synthases, CerS2 in a GATA-1-dependent manner and that the blockade of CCR5 signalling increased CerS2 and reduced TCR nanoclustering. Finally, the analysis of

TCR nanoclustering in activated human CD4⁺ T cells from healthy CCR5 WT or ccr5Δ32 homozygous donors confirmed both the *in vivo* and *in vitro* data obtained in the mouse model.

The results are very interesting and relevant for understanding the role of CCR5 signalling in the regulation of the functional fitness of memory cells. Moreover, several *in vivo* and *in vitro* experimental systems as well mathematical models have been used for demonstrating an important role of CCR5-triggered transcription programs in regulating ceramide metabolism and related TCR nanoclustering in both human and mouse CD4⁺ memory T cells. At a pathological level, these findings also suggest a novel potential therapeutic target for enhancing memory T cell response in ccr5Δ32 homozygous donors and at the same time reinforce the notion of the risks of genetic ablation of CCR5 to prevent HIV infection.

Minor points:

Figures and supplementary figures.

It would be useful to the readers to add the mean values of the comparative groups in the legends of each figure.

Figure 2. Panel A. In addition to the representative histograms, dot plots of the gating strategy used for analysing both cytokines and memory markers in antigen-experienced CD4⁺ T cell would be useful and should be added. Panel B Mean values should be added in the legends

Figure 3. Panels A-C. It would be useful to specify in the legend that images and data were from EM. If possible, it would be clearer to the readers if all EM images are cropped together in one panel and the relative distribution histograms in separate panels. Panel F Mean values should be added in the legend.

Figure 4. It would be useful to specify in the legend that images and data were from EM.

Figure 5. The figure has been marked as 4. Mean values should be added in the legend of panels A, E, F, G, M and N.

Figure 6. Panels D and G. Mean values should be added in the legend.

Figure 7: Panel A. It would be useful to specify in the legend that images and data in the insets were from EM. Panel C. Mean values should be added in the legend.

Referee #2:

The network of chemokines and their cognate receptors are essential for orchestrating cell migration. The chemokine receptor CCR5 is expressed on several innate and adaptive immune cell subtypes, including effector and memory T cells. Besides its role in guiding cell migration, CCR5 is well known for its HIV-1 co-receptor function. Moreover, CCR5 acts as T cell co-stimulatory molecule and is involved in T cell differentiation. As correctly introduced in the present manuscript, a single but seminal paper more than a decade ago provided evidence that CCR5 guides naïve CD8 T cells to antigen-bearing DC/CD4 T cell complexes and that interfering with this recruitment/interaction seems to reduce CD8 T cell memory generation through an unknown mechanism. Here, Martin-Leal and colleagues identify that CCR5 deficiency impairs CD4 T cell memory responses (in the OVA/OTII model) determined by the % of IFN γ producing cells upon antigen restimulation in a cell-autonomous fashion and without affecting the total number of CD4 memory cells, nor the percentage of CD4 TEM. Moreover, the authors identified that the TCR forms nanoclusters in antigen-experienced CD4 T cells, whereas TCR nanoclustering is impaired in naïve and CCR5 deficient CD4 T cells. This TCR nanoclustering depended on CCR5-mediated G-protein

signaling, but non on the CCR5 co-stimulatory activity. Notably, the authors found that CCR5 deficient T cells express higher levels of ceramide synthases and hence possess increased ceramide levels, but normal cholesterol levels. As interfering with ceramide levels in CD4 T cells affected TCR nanoclustering clearly suggest that CCR5 signaling modulated TCR nanoclustering in antigen-experienced CD4 T cells through modulating ceramide levels. Importantly, human individuals expressing a defective variant of CCR5, CCR5d32, show reduced TCR nanoclusters, which could explain why ccr5d32 homocytous individuals are more prone to fatal infections.

Overall, this is a highly interesting study. However, several drawbacks detract from the importance of the work.

1. TCR nanoclustering is measured solely by immunogold labelling and EM, which has its limitation that not all receptors are labelled. Importantly, EM pictures (Fig 3B,C,G, 4A,B,C,7A) show that the most gold particles are found in clusters, only a small fraction is seen as individual particle in wild-type cells. Clearly discrepant to this, quantification of gold particles per cluster shown below the EM picture indicate that most 'clusters' include a single gold particle. This discrepancy must be solved as these data are central for the entire study. In addition, although many gold particles were counted, they all derive from a very small number of individual cell (<10!). What is the percentage of nanoclusters/cell in the population? Is there a difference in effector memory versus central memory T cells? Conceptually, as only a minor fraction of TCR form nanoclusters on an individual cell, how can this affect memory development?

To corroborate TCR nanoclustering, alternative methods, such as proximity ligation or antibody-mediated FRET on primary naïve versus memory cells, or FRET/BRET on cell lines with altered ceramide levels shall be used.

2. Functionality of CCR5 was determined by Transwell migration assay. Data shown in Fig. S1 are not convincing, as barely any migration is shown. According to the method section, migration was measured in response to 100mM (miliM!) of CCL4. At this chemokine concentration no T cell is migrating. For T cells, maximal chemotactic response is seen at 1-10 nM!

3. Receptor clustering is favored particularly by fully saturated lipids. It is intriguing to see that the partially unsaturated C24:1 ceramide species show highest difference between wild-type and CCR5 deficient mouse T cells. Notably, this is different in human cells. Is there an explanation for this?

4. The authors claim that CCR5 signaling limits GATA1-induced transcription of ceramide synthases (e.g. abstract) without providing experimental evidence. They provide only correlative evidence. Conditional deletion of GATA1 or its pharmacological inhibition is required to conclude this important mechanistic insight.

5. Do CCR5 molecules also form nanoclusters? And does signaling through CCR5 nanoclusters serve as feedback loop?

6. Fig 4H: PTx effect is more pronounced at 10⁻² ug/ml than at 10⁻¹. Why?

7. Page 12: authors claim that 'SMase treatment reduced the number of high valency TCR clusters'. But no such data is shown.

8. Fig. 6: data on CerS2 silencing is not convincing and additional evidence must be provided.

Minor points.

- Fig.1 J,K: some bars are colorcoded in red, but the colorcode is not specified in the chart and figure legend. Additional, unexplained colorcoding is also seen in Fig 3D,F, 5B,E,N, 6I

- There are two Fig 4, but no Fig 5.

- Fig 5. Panel letters must be in capital

Point-by-point reply

EMBOJ-2020-104749

Reviewer #1

We thank the reviewer for analyzing our study and for his/her helpful comments. We address the points raised as follows:

1. Figures and supplementary figures. It would be useful to the readers to add the mean values of the comparative groups in the legends of each figure.

Although we understand the reviewer's point, we think that adding all these data could create more confusion than help. For instance, in Figure 1 alone, we would have to add 12 data items, with mean \pm SEM for the different paired comparisons. In addition, we would have to add all mean values for each Ig subclass in Fig. 1J and K (i.e., 24 comparisons). In our view, addition of this much data will create reader confusion while providing no relevant information, as all graphs show individual values as well as mean and SEM.

2. Figure 2. Panel A. In addition to the representative histograms, dot plots of the gating strategy used for analysing both cytokines and memory markers in antigen-experienced CD4+ T cell would be useful and should be added. Panel B Mean values should be added in the legends.

We concur with the reviewer's opinion, and have added contour plots of the gating strategy used for memory markers as an expanded view figure (Figure EV1). We would like to clarify that cytokines were determined by ELISA in the supernatant, and there are thus no associated gating strategies.

3. Figure 3. Panels A-C. It would be useful to specify in the legend that images and data were from EM. If possible, it would be clearer to the readers if all EM images are cropped together in one panel and the relative distribution histograms in separate panels. Panel F Mean values should be added in the legend.

We have clarified which images and data were obtained from EM. We have also added an expanded view figure (Figure EV2) showing the entire surface replica of a representative cell, with the appearance of TCR clusters and monomers. We appreciate the reviewer's suggestion regarding figure organization, and tried arranging the panels as suggested; nonetheless, we feel the reader benefits by simultaneously comparing the EM images and their quantification.

4. Figure 4. It would be useful to specify in the legend that images and data were from EM.

We have clarified which images and data were obtained from EM.

5. Figure 5. The figure has been marked as 4. Mean values should be added in the legend of panels A, E, F, G, M and N.

We apologize for the labeling mistake, now corrected. We have capitalized the panel letters.

6. Figure 6. Panels D and G. Mean values should be added in the legend.

As above, the inclusion of mean values in the figure legend could generate confusion, and exceed the permitted length of the legend.

7. Panel A. It would be useful to specify in the legend that images and data in the insets were from EM. Panel C. Mean values should be added in the legend.

We have clarified which images and data were obtained from EM.

Reviewer #2

We thank the reviewer for analyzing our study and for his/her helpful comments. We have addressed the criticisms as follows:

1a. TCR nanoclustering is measured solely by immunogold labelling and EM, which has its limitation that not all receptors are labelled.... To corroborate TCR nanoclustering, alternative methods, such as proximity ligation or antibody-mediated FRET on primary naïve versus memory cells, or FRET/BRET on cell lines with altered ceramide levels shall be used.

In the Discussion, we recognized the limitations of EM. We would nonetheless emphasize that our aim is not to determine the absolute number of TCR nanoclusters in a cell, but to compare the relative number of nanoclusters of a given size in two experimental conditions.

We also want to clarify that EM is not the only method used to analyze differences in nanoclustering, but that Blue-Native PAGE was used as well to validate the EM results between WT and CCR5-KO cells (Fig. 3F). There are several reasons we chose BN-PAGE as a complementary approach to verify differential TCR nanoclustering between WT and CCR5-KO cells. The principal motive is that this method is based on a different conceptual principle; that is, whereas EM relies on high-resolution imaging, BN-PAGE is essentially biochemical. A second reason is that many of the image-based techniques require adhering cells to a substrate, which in turn can induce adhesion-dependent signals that affect TCR distribution. This handicap does not apply to EM, since TCR nanoclusters are analyzed only at the apical side of the cell, nor to BN-PAGE, which examines TCRs at the whole cell level in non-attached cells. Finally, all the microscopic techniques are based on the use of antibodies or overexpression of modified receptors. In some cases, these antibodies must be single-labeled anti-TCR F(ab), which might have very low affinity. In contrast, BN-PAGE is independent of antibody use and does not require TCR overexpression.

We nonetheless appreciate the reviewer's suggestions, and indeed some of the coauthors are working to establish PLA and FRET analysis methods to study TCR nanoclustering. This is not an easy task, but requires considerable technical development probably outside the scope of this manuscript, in which we aim to understand how CCR5 influences CD4⁺ T cell memory.

1b. ...Importantly, EM pictures (Fig 3B,C,G, 4A,B,C,7A) show that the most gold particles are found in clusters, only a small fraction is seen as individual particle in wild-type cells. Clearly discrepant to this, quantification of gold particles per cluster shown below the EM picture indicate that most 'clusters' include a single gold particle. This discrepancy must be solved as these data are central for the entire study.

As the reviewer states, quantification of the EM images clearly shows that there is a big fraction of the TCR molecules as monomeric by this technique. It is nevertheless difficult to find images that combine monomeric and multimeric TCR in this proportion. We have tried to present representative images showing the differences in nanocluster valency between the experimental groups, as we think this is more informative than showing single TCR molecules. We have nonetheless added an expanded view figure (Figure EV2) of the whole surface replica of a representative cell, to show TCR monomer and nanocluster distribution on the cell surface.

1c. ... In addition, although many gold particles were counted, they all derive from a very small number of individual cell (<10!).

The cells used for counting were those in perfect conditions. We mention that few cell replicas maintain membrane integrity after the freezing and cryofracture steps required for

EM. Despite the small number of cells examined, the total number of gold particles counted is in the order of thousands or tens of thousands, sufficient for statistical purposes.

1d. What is the percentage of nanoclusters/cell in the population?

The percentage of nanoclusters/cell obviously varies depending on cell genotype and activation/differentiation status. We include a supplementary table showing the mean \pm SEM of TCR nanoclusters per cell. These data suggest that cell-to-cell variation is relatively low in TCR molecules that form nanoclusters in a given condition.

1e. Is there a difference in effector memory versus central memory T cells? Conceptually, as only a minor fraction of TCR form nanoclusters on an individual cell, how can this affect memory development?

Our data suggest that CCR5 does not affect memory development; indeed, the frequency of effector and central memory T cells is equivalent between WT and CCR5^{-/-} mice (Fig. 1A, C and D). Instead, our results suggest that CCR5 affects the ability of CD4⁺ T cells to be activated at low antigen concentration (Fig. 1E), probably due to impaired TCR nanoclustering. Although relatively few nanoclusters are present, these are the TCR activated preferentially at low Ag concentration (see Schamel et al., JEM 2005). We agree with the reviewer that to analyze whether CCR5 effect is similar in central and effector memory is a very interesting question. The purification kit we used to analyze TCR nanoclustering in memory cells from NIP-OVA-immunized mice (Fig. 3G) purifies mainly effector memory CD4⁺ T cells.

3. Functionality of CCR5 was determined by Transwell migration assay. Data shown in Fig. S1 are not convincing, as barely any migration is shown. According to the method section, migration was measured in response to 100mM (miliM!) of CCL4. At this chemokine concentration no T cell is migrating. For T cells, maximal chemotactic response is seen at 1-10 nM!

The chemotactic activity was analyzed using CCL4 at 100 nM; we apologize for the mistake. We recognize that this is an unusually high dose to see chemotaxis, and indeed in preliminary experiments we used lower doses. However, the best results were obtained at 100 nM. A possible explanation is that we assayed antigen-experienced T cells. In our experience, activated T cells show less chemotaxis than naïve T cells, whereas in contrast their migration is increased in the absence of chemoattractant; for this reason, the migration index was low. We have replaced the graph, showing the same data expressed as a percentage of migration. In addition, we include new data showing the effect of CCL4 on Ca²⁺ flux.

3. Receptor clustering is favored particularly by fully saturated lipids. It is intriguing to see that the partially unsaturated C24:1 ceramide species show highest difference between wild-type and CCR5 deficient mouse T cells. Notably, this is different in human cells. Is there an explanation for this?

The different ceramidome in mice and human cells could depend on many factors, such as cell status at the time of harvest, the profile of enzymes involved in sphingolipid metabolism, or the lipid composition of the diet. Membrane phase separation properties of ceramides are dependent not only on the saturation but also the length of the acyl chain. The longer the acyl chain, the greater the stiffness of the lipid bilayer. The differences in the membrane phase separation properties of C24:0 and C24:1 ceramides might be subtle, particularly in an environment with other long-chain saturated and unsaturated ceramides, as well as dihydroceramides. C24:1 is the most abundant ceramide in mouse WT and CCR5^{-/-} T cells. However, the largest differences associated to CCR5 in mouse cells were observed for C20:0 and C22:0 ceramides. A discussion of these data has been included.

4. *The authors claim that CCR5 signaling limits GATA1-induced transcription of ceramide synthases (e.g. abstract) without providing experimental evidence. They provide only correlative evidence. Conditional deletion of GATA1 or its pharmacological inhibition is required to conclude this important mechanistic insight.*

We concur with the reviewer that the evidence provided is correlative. Nevertheless, generation of a conditional GATA1 knockout mouse, and later a quadruple transgenic (GATA1^{f/f}, CD4^{Cre}, CCR5^{-/-}, OT-II^{tg}) mouse strain, is a full research project in itself. We are not aware of available GATA1 inhibitors, and our past experience with pharmacological inhibitors for other transcription factors has been very disappointing. We have thus modified the text to soften our conclusions on GATA1-mediated control of CerS2. We hope this solution will satisfy the reviewer.

5. *Do CCR5 molecules also form nanoclusters? And does signaling through CCR5 nanoclusters serve as feedback loop?*

It has been demonstrated that chemokine receptors, including CCR5, can form dimers and multimers; they can therefore form nanoclusters. The dimerization process can indeed change the signaling properties of the chemokine receptors, but certainly it is difficult to ascertain whether these chemokine receptor nanoclusters will boost or prevent TCR nanoclustering.

We tried to visualize CCR5 and TCR nanoclusters simultaneously, using gold particles of different sizes. However, we were unable to find an antibody that recognizes mouse CCR5 with minimal specificity criteria; all antibodies tested (from several commercial sources) cross-reacted with CCR5^{-/-} T cells.

We have slightly extended our discussion of CCR5 nanoclusters and the potential feedback in TCR nanoclustering.

6. *Fig 4H: PTx effect is more pronounced at 10⁻² ug/ml than at 10⁻¹. Why?*

Although we performed preliminary experiments to determine the maximal PTx dose that does not cause cell death (as determined by annexin V staining), it is possible that the highest dose used in this experiment might slightly affect cell fitness. As a result, data dispersion is higher in PTx-treated cells at 10⁻¹ µg/ml than at 10⁻² µg/ml, giving the impression that the effect is reduced at the highest PTx concentration. We prefer to keep these data, however, since we have no clear proof that PTx at 10⁻¹ µg/ml causes cell death at this concentration. In any case, the differences between 10⁻¹ and 10⁻² µg/ml data points are not statistically significant.

7. *Page 12: authors claim that 'SMase treatment reduced the number of high valency TCR clusters'. But no such data is shown.*

These data were shown in Fig. 6E.

8. *Fig. 6: data on CerS2 silencing is not convincing and additional evidence must be provided.*

We understand that the criticism refers to the increase in CD69 levels observed after restimulation of shCerS2-silenced TAK779-treated 2B4 cells. The reviewer should consider that 2B4 is a T cell hybridoma that constitutively expresses very high CD25 and IL-2 levels. Likewise, 2B4 cell proliferation is independent of TCR activation. The only activation marker we found to be regulated by TCR activation was CD69. We also performed CerS2 silencing in primary CCR5^{-/-} T cells (Supplemental Fig. S9); although CerS2 was not efficiently silenced, we found that both proliferation and IL-2 secretion tended to be higher in shCerS2- than in shCtrl-transduced CCR5^{-/-} T cells. Indeed, the differences in proliferation were statistically significant at one antigen dose used. We thus consider that the CD69 data for 2B4 cells, and the proliferation and IL-2 data obtained in primary cells suggest that CerS2 silencing increases the antigen response by cells lacking (or with inhibited) CCR5 signaling.

Minor points.

- *Fig. 1 J,K: some bars are colorcoded in red, but the colorcode is not specified in the chart and figure legend. Additional, unexplained colorcoding is also seen in Fig 3D,F, 5B,E,N, 6I*

Color was lost in some parts of the figures during PDF file conversion. We apologize for this oversight in our review of the converted files.

- *There are two Fig 4, but no Fig 5.*

We apologize for the mistake, which has been corrected.

- *Fig 5. Panel letters must be in capital.*

Panel letters have been capitalized.

Dear Santos,

Thank you for submitting your revised manuscript. Your study has now been seen referee #2. The referee appreciates the revised version and has just a few remaining comments - please see below. I am therefore very pleased to let you know that we can accept the manuscript for publication here. Before sending you the formal acceptance letter there are just a few issues to sort out. You can use the link below to upload the revised version.

- We need 5 keywords
- Appendix Figure S1 is missing the word 'Appendix'.
- Our publisher has also done their pre-publication check on your manuscript. I will send you their comments in a separate email
- We also require a Data Availability Section. As far I can see you have no data to deposit in databases and therefore please state something like This study includes no data deposited in external repositories.

That should be all - Congratulations on a nice study

with best wishes

Karin

Karin Dumstrei, PhD
Senior Editor
The EMBO Journal

The revision must be submitted online within 90 days; please click on the link below to submit the revision online before 9th Aug 2020.

Link Not Available

Referee #2:

The authors have substantially revised their manuscript and adequately addressed the majority of the raised concerns.

The following minor changes are required prior accepting the manuscript for publication:

-Page 8/ Fig. EV2: please specify whether the image derive from a naive cell or a lymphoblast. Ideally, a representative image of a wild-type lymphoblast should be set against an image derived from a CCR5-deficient lymphoblast.

- Page 8: Not the percentage, but the effective number of TCR nanoclusters is indicated in Appendix Table S1. The text must be corrected. The total number of cells used to count the nanoclusters must be specified in table S1, as well as a definition of how many TCR within a certain area (presumably >4) are considered to form a nanocluster must be specified.

Figure 3A,B,C,G: for the sake of consistency and as requested in the previous review, bars representing CCR5^{-/-} cells shall be color-coded in red, not black, as for the other figures and panels.

Page 13-14: SMase treatment and CerS2: the authors shall not only refer to Fig 6E & I, but also to Appendix Table S1 where TCR nanoclusters have been enumerated. Similarly TCR clustering in human T cells shall not only refer to Fig 7a, but also to Appendix Table S1 (page 15).

Figure 7A: for the sake of consistency, bars representing ccr5D32 shall be color-coded in light red, not black, as for all other panels in Fig 7.

Dear Santos,

Thank you for submitting your revised manuscript. I have now looked at everything and all looks good. I am therefore very pleased to accept the manuscript for publication here.

Congratulations on a nice study!

with best wishes

Karin

Karin Dumstrei, PhD
Senior Editor
The EMBO Journal

Please note that it is EMBO Journal policy for the transcript of the editorial process (containing referee reports and your response letter) to be published as an online supplement to each paper. If you do NOT want this, you will need to inform the Editorial Office via email immediately. More information is available here: http://emboj.embopress.org/about#Transparent_Process

Your manuscript will be processed for publication in the journal by EMBO Press. Manuscripts in the PDF and electronic editions of The EMBO Journal will be copy edited, and you will be provided with page proofs prior to publication. Please note that supplementary information is not included in the proofs.

Should you be planning a Press Release on your article, please get in contact with embojournal@wiley.com as early as possible, in order to coordinate publication and release dates.

If you have any questions, please do not hesitate to call or email the Editorial Office. Thank you for your contribution to The EMBO Journal.

** Click here to be directed to your login page: <http://emboj.msubmit.net>

Corresponding Author Name: Santos Mañes

Manuscript Number: EMBOJ-2020-104749